# Cell-specific responses to the cytokine TGFβ are determined by variability in protein levels

Jette Strasen[1,†], Uddipan Sarma[2,†], Marcel Jentsch[1,3,†], Stefan Bohn[3], Caibin Sheng[1,3], Daniel Horbelt[4], Petra Knaus[4], Stefan Legewie[2,*] (ID) & Alexander Loewer[1,3,**] (ID)

## Abstract

The cytokine TGFβ provides important information during embryonic development, adult tissue homeostasis, and regeneration. Alterations in the cellular response to TGFβ are involved in severe human diseases. To understand how cells encode the extracellular input and transmit its information to elicit appropriate responses, we acquired quantitative time-resolved measurements of pathway activation at the single-cell level. We established dynamic time warping to quantitatively compare signaling dynamics of thousands of individual cells and described heterogeneous single-cell responses by mathematical modeling. Our combined experimental and theoretical study revealed that the response to a given dose of TGFβ is determined cell specifically by the levels of defined signaling proteins. This heterogeneity in signaling protein expression leads to decomposition of cells into classes with qualitatively distinct signaling dynamics and phenotypic outcome. Negative feedback regulators promote heterogeneous signaling, as a SMAD7 knock-out specifically affected the signal duration in a subpopulation of cells. Taken together, we propose a quantitative framework that allows predicting and testing sources of cellular signaling heterogeneity.

**Keywords** cellular heterogeneity; mathematical modeling; signaling dynamics; single-cell analysis; TGFβ-SMAD signaling

**Subject Categories** Quantitative Biology & Dynamical Systems; Signal Transduction

**Mol Syst Biol. (2018) 14: e7733**

## Introduction

Cells sense their surrounding using cell-surface receptors and signaling pathways that transmit the corresponding information from the cell membrane to the nucleus. Cellular signaling is able to quantitatively respond to fine-grained inputs, for example, during development, when morphogens precisely determine cell fates according to spatial localization (Gurdon et al, 1998). However, it remains poorly understood how mammalian cells encode and decode quantitative information about extracellular inputs. Recent studies have shown that temporal dynamics of pathway activity can contribute to specific information processing and determine cellular responses (Purvis & Lahav, 2013). To measure dynamics of cellular signaling, live-cell imaging of fluorescent reporters emerged as a powerful approach (Spiller et al, 2010). In addition to providing unparalleled temporal resolution, it allowed to follow signaling in thousands of individual cells over time. This revealed that genetically identical cells frequently respond in different ways to the same external stimulus. For p53, TNF-α, and NF-κB signaling, it has been demonstrated that due to non-genetic heterogeneity, the signaling dynamics of each individual cell determine the phenotypic response to extracellular stimulation (Geva-Zatorsky et al, 2006; Ashall et al, 2009; Spencer et al, 2009; Tay et al, 2010; Purvis et al, 2012; Lee et al, 2014).

Further studies confirmed that precise information transmission is in general limited by non-genetic heterogeneity, leading to differences in differentiation programs (Chang et al, 2008; Goolam et al, 2016), drug resistance (Cohen et al, 2008; Sharma et al, 2010; Paek et al, 2016), and viral pathogenesis (Weinberger et al, 2005). Heterogeneity in signaling emerges from various molecular sources including cell cycle stage, external influences such as the microenvironment, or stochastic intracellular events (Loewer & Lahav, 2011; Snijder & Pelkmans, 2011). Stochasticity may arise due to the stochastic dynamics of biochemical reactions in a signaling pathway (Rand et al, 2012), or from noise in gene expression that leads to cell-to-cell variability in the concentrations of signaling proteins (Feinerman et al, 2008). We therefore need a quantitative time-resolved characterization of mammalian signaling systems at the single-cell level to understand and predict how each individual cell will respond to a given extracellular input.

A crucial extracellular input during embryonic development, adult tissue homeostasis, and regeneration is the cytokine TGFβ (Schmierer & Hill, 2007; Heldin et al, 2009). TGFβ stimulation

---

1 Berlin Institute for Medical Systems Biology, Max Delbrueck Center in the Helmholtz Association, Berlin, Germany
2 Institute of Molecular Biology (IMB), Mainz, Germany
3 Department of Biology, Technische Universität Darmstadt, Darmstadt, Germany
4 Institute for Chemistry and Biochemistry, Freie Universität Berlin, Berlin, Germany
 *Corresponding author. Tel: +49 6131 39 21430; E-mail: s.legewie@imb-mainz.de
 **Corresponding author. Tel: +49 6151 16 28060; E-mail: loewer@bio.tu-darmstadt.de
 †These authors contributed equally to this work

prevents uncontrolled tissue growth by inducing cell cycle arrest and apoptosis and can trigger epithelial-to-mesenchymal transition (EMT), a conversion of adherent epithelial cells into a migratory, mesenchymal phenotype (Gonzalez & Medici, 2014). TGFβ signaling is dysregulated during pathological conditions such as organ fibrosis and cancer. In tumorigenesis, the pathway plays a dual role: Many early-stage tumors evade the tumor-suppressive, cell cycle inhibitory role of TGFβ, whereas its EMT-promoting function frequently induces metastasis of late-stage tumors (Ikushima & Miyazono, 2010). Thus, a specificity switch from one cellular response to another can occur in TGFβ signaling. The underlying molecular changes are currently unclear and may involve changes in the expression of transcription factors (Mullen et al, 2011) and signaling proteins (Piek et al, 2001), or altered temporal dynamics of the pathway (Nicolás & Hill, 2003).

TGFβ initiates signaling through binding to and activation of its serine/threonine kinase transmembrane receptors (TGFβRI and TGFβRII). Ligand binding triggers receptor-mediated phosphorylation of SMAD2/3, which then heterotrimerize with SMAD4, translocate to the nucleus and bind to target gene promoters for transcriptional regulation (Feng & Derynck, 2005). This results in gene expression changes including the downregulation of classical epithelial and cell cycle genes and upregulation of mesenchymal markers (Massagué, 2005). Additionally, TGFβ target genes include negative feedback regulators of the pathway.

Previous experimental and theoretical studies quantitatively characterized the mechanisms shaping the temporal dynamics of SMAD signaling (Clarke & Liu, 2008; Schmierer et al, 2008; Zi et al, 2012). One important mechanism that limits the duration of the signal is the depletion of extracellular TGFβ due to internalization of receptor–ligand complexes, followed by lysosomal TGFβ degradation (Clarke et al, 2009; Zi et al, 2011). Internalization of signaling complexes may also deplete TGFβ receptors from the cell membrane (Vizan et al, 2013), thereby contributing to a refractory period in which cells are insensitive to further TGFβ stimuli (Vizan et al, 2013; Sorre et al, 2014). In the nucleus, phosphatases such as PPM1A revert the phosphorylation of SMAD2/3 and facilitate their export to the cytoplasm (Lin et al, 2006). Finally, transcriptional feedbacks acting at multiple levels including receptor deactivation (Valdimarsdottir et al, 2006; Wegner et al, 2012) or SMAD dephosphorylation (Wang et al, 2014a) contribute to signal termination.

Previous quantitative analyses of SMAD signaling mainly focused on average behavior of a cell population at defined time points, whereas the long-term response at the level of individual cells is much less well characterized. Recent studies revealed that SMAD2-SMAD4 complex formation and nuclear translocation of fluorescently labeled SMAD proteins occur with pronounced cell-to-cell variability (Warmflash et al, 2012; Zieba et al, 2012). Heterogeneous signaling behavior at selected time points post-stimulation was shown to be partially related to cell density and cell cycle stage (Zieba et al, 2012). However, to understand how TGFβ signaling elicits defined responses in a cell-specific and concentration-dependent manner, we need to systematically characterize its dynamics on the single-cell level and integrate experimental measurements with quantitative mathematical models of the underlying molecular interactions. This would allow us to predict how individual cells react to a given input and to design targeted perturbations of the pathway to exploit its role in health and disease.

To this end, we combined live-cell imaging of fluorescent SMAD2 and SMAD4 fusion proteins with automated image analyses to quantitatively characterize long-term dynamics of TGFβ signaling in individual cells. Based on clustering of thousands of time courses, we identified six cellular subpopulations with qualitatively distinct signaling behavior and concluded that the phenotypic response of an individual cell is determined by the temporal dynamics of SMAD nuclear translocation. We described the dynamics of these subpopulations and of the complete heterogeneous cell population using a quantitative modeling approach. This theoretical and experimental approach revealed that heterogeneity in signaling arises from varying levels of signaling proteins. A CRISPR/Cas9-mediated knock-out of SMAD7 confirmed our model prediction that a major part of the observed heterogeneity can be attributed to fluctuations in feedback proteins. Taken together, we present a framework to characterize the response of cellular subpopulations to external cues and to quantitatively model the underlying molecular mechanisms of signaling heterogeneity. Furthermore, our results place the cell-specific temporal dynamics of SMAD signaling as an important determinant of the variegated cell fates elicited by TGFβ stimuli.

# Results

## Quantitative imaging of SMAD nuclear translocation at the single-cell level

A key step in TGFβ signaling is the translocation of SMAD transcription factor complexes from the cytoplasm to the nucleus. To monitor this translocation event in individual cells with high temporal and spatial resolution, we established a live-cell reporter system based on the breast epithelial cell line MCF10A, an established model for TGFβ signaling (Zhang et al, 2014). To this end, we generated a stable clonal cell line expressing a YFP-SMAD2 fusion protein under the control of a constitutive promoter as well as histone H2B-CFP as a nuclear marker (Fig 1A). Western blot analysis revealed that the amount of SMAD2-YFP fusion protein corresponds to approximately 50% of the endogenous SMAD2 protein (Fig 1B). We validated that this overexpression did not perturb the dynamics of SMAD2 signaling by monitoring TGFβ1-induced phosphorylation of endogenous SMAD2 in the parental and reporter cell lines (Figs 1C and EV1A). Furthermore, qPCR analysis revealed that the induction of well-characterized SMAD target genes in response to TGFβ1 stimulation remained essentially unchanged (Fig EV1B).

To measure SMAD2-YFP translocation in living cells, we performed time-lapse imaging over a 24-h time interval after a saturating TGFβ1 stimulus. In the example cell shown, SMAD2 predominantly located to the cytoplasm in the absence of TGFβ1 as expected and strongly accumulated in the nucleus within 1 h of stimulation (Fig 1D). After this initial response, SMAD2 relocalized to the cytoplasm, before it accumulated in the nucleus again about 5 h post-stimulation. Nuclear SMAD2 then remained elevated at varying levels throughout the experiment. As we aimed to compare SMAD2 dynamics in hundreds of cells, we employed automated image analysis to quantify the nuclear and cytoplasmic SMAD2 concentrations and expressed the signaling pathway activity as their ratio (nuc/cyt

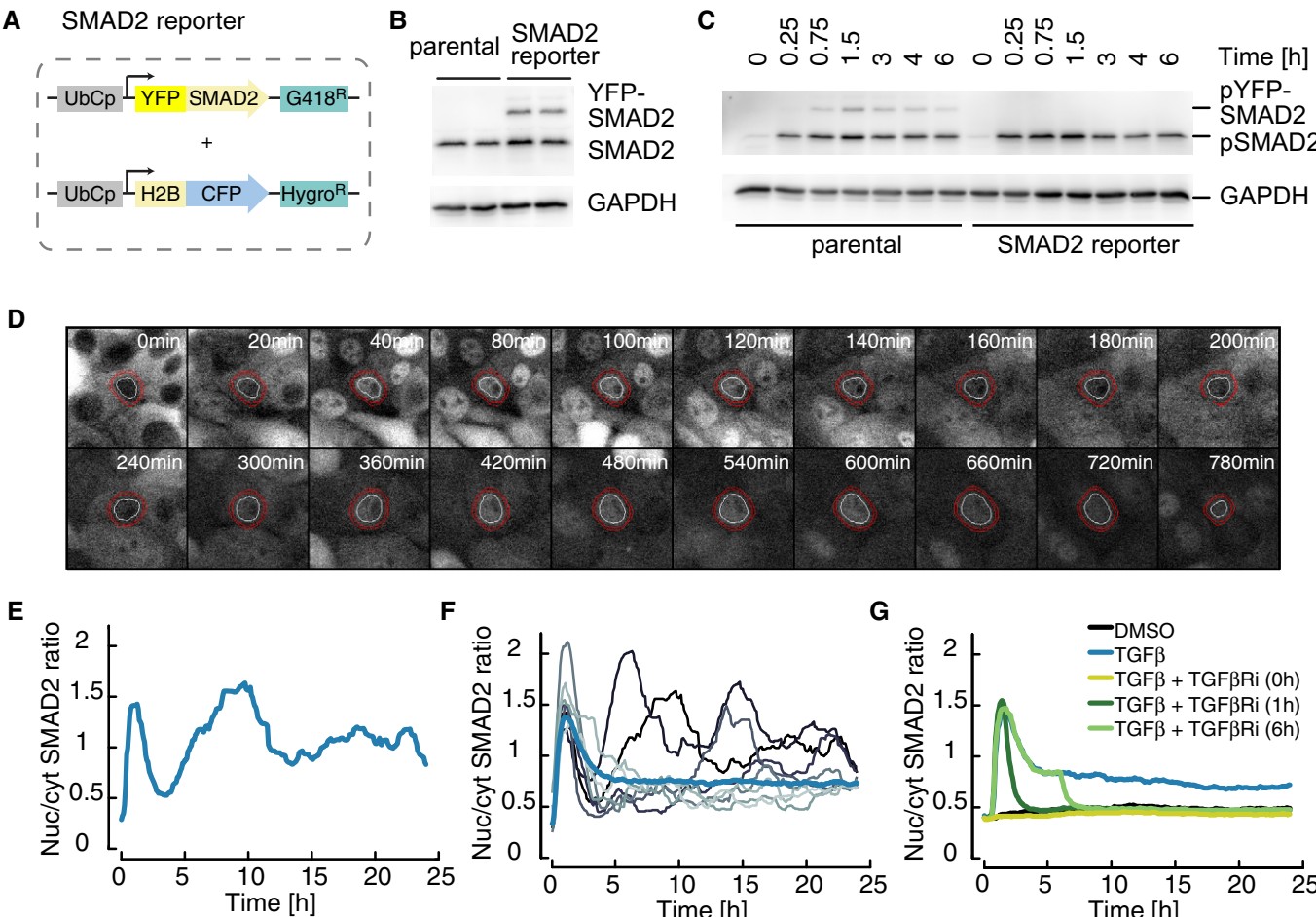

**Figure 1.  Dynamics and variability of SMAD2 signaling in single cells.**

A     Fluorescent reporter system to measure SMAD signaling dynamics in individual cells. SMAD2 was fused to the yellow fluorescent protein mVenus (YFP) under the control of the human ubiquitin C promoter (UbCp) with the selection marker G418 (Geneticin). As a nuclear marker, histone 2B (H2B) was fused to the cyan fluorescent protein mCerulean (CFP) under the control of UbCp with the selection marker hygromycin.

B     Western blot analysis of endogenous and YFP-tagged SMAD2 in a stable clonal reporter cell line and the corresponding parental cell line. Cells were stimulated with 100 pM TGFβ1 and analyzed after 3 h. GAPDH was used as a loading control.

C     Western blot analysis of SMAD2 activation in SMAD2-YFP reporter and parental MCF10A cells. Cells were stimulated with 100 pM TGFβ1, and SMAD2 phosphorylation was analyzed at indicated time points. GAPDH was used as a loading control.

D, E   Live-cell time-lapse microscopy images of MCF10A cells expressing SMAD2-YFP following treatment with 100 pM TGFβ1 (D). White circles indicate the segmented nucleus, and the estimated cytoplasmic area is represented by red annuli. The indicated cell was tracked over 24 h and the corresponding nuclear-to-cytoplasmic (nuc/cyt) SMAD2-YFP ratio plotted over time (E).

F     Time-resolved analysis of the SMAD2 nuclear to cytoplasmic localization for eight individual cells (thin lines) compared to the median nuc/cyt SMAD2 ratio of the entire population (thick line) upon stimulation with 100 pM TGFβ1. See Appendix Table S1 for number of cells analyzed.

G     Median nuc/cyt SMAD2 ratio for reporter cells stimulated with 100 pM TGFβ1 and treated with TGBβRI kinase inhibitor (SB431542) at indicated time points. At all time points, SMAD2 nuclear translocation was dependent on TGFβ receptor activity. See Appendix Table S1 for number of cells analyzed.

ratio, Figs 1E and EV1C–F, Appendix Fig S1 and Appendix II.A and II.B). This measure was robust against correlated fluctuations due to heterogeneity of transgene expression or measurement aberrations such as photobleaching and reproducible between biological replicates (Fig EV1G). We validated that changes in the nuc/cyt ratio of SMAD2 reflect the kinetics of receptor-mediated phosphorylation of endogenous SMAD2 (Fig EV1H and I). When cells divided during the duration of the experiment, we only followed one of the daughter cells and merged mother and daughter trajectories before and after division (see Appendix II.A).

Using this approach, we observed substantial heterogeneity in the response to the saturating stimulus (Fig 1F). Most cells showed nuclear SMAD2 accumulation shortly after the initial stimulus. However, some cells immediately adapted to a low signaling plateau afterward, whereas others were characterized by renewed nuclear translocation of SMAD2. The average response of all cells in the population revealed signaling dynamics similar to biochemical measurements of cell populations in previously published studies (Inman *et al*, 2002; Clarke *et al*, 2009; Zi *et al*, 2011; Vizan *et al*, 2013). Importantly, nuclear translocation of SMAD2 was dependent

on TGFβ receptor activity at all time points, as signaling was rapidly and synchronously terminated in all cells by the specific inhibitor SB431542 (Fig 1G; Inman *et al*, 2002). We observed comparable heterogeneous dynamics for SMAD4 nuclear translocation using a similarly engineered and validated reporter cell line (Appendix Fig S2).

### Dynamic features of SMAD signaling encode phenotypic responses

Next, we investigated whether heterogeneous signaling was limited to saturating TGFβ1 concentrations or a characteristic feature of the pathway at all stimulus levels. We treated cells with varying TGFβ1 doses and quantified SMAD2 localization over a 24-h period. Interestingly, we again observed pronounced cell-to-cell variability (Fig 2A). At low stimulation levels, cells either showed almost no response to the input or transient nuclear SMAD2 accumulation over the first 5 h. At higher TGFβ1 concentrations, most cells showed an initial response to the input. However, the extent and duration of renewed nuclear SMAD2 translocation at later time points were highly variable: A single-cell response to 25 pM TGFβ1 could be transient and of limited amplitude, resembling trajectories typically observed upon stimulation with 5 pM TGFβ1 (Fig 2A). In essence, dynamic signaling responses were overlapping between input levels and therefore only partially determined by the strength of the extracellular stimulus.

TGFβ is known to control cell fate in a dose-dependent manner (Schmierer & Hill, 2007). Accordingly, we find that changing the TGFβ1 stimulus alters the median SMAD2 response and expression levels of selected target genes in cell populations (Figs 2B and EV2A and B). How does the SMAD pathway encode dose-dependent information despite the strong cellular heterogeneity observed in our single-cell measurements? We hypothesized that phenotypic responses are determined by the individual pattern of SMAD translocation in a given cell rather than by the amount of ligand applied to a population. To quantify pair-wise differences between single-cell time courses, we used dynamic time warping (DTW), a method for non-linear alignment in the time domain, which is frequently employed in speech analysis (Sakoe & Chiba, 1978). Compared to simpler metrics such as Euclidean distance, DTW is more robust against distortions in the time domain and therefore emphasizes dynamic patterns while preserving differences in amplitudes (Fig EV2C). To improve its applicability to biological systems, we modified DTW by introducing an elastic constraint on stretching a given time series (cDTW, see Appendix Fig S3 and Appendix II.C for more information on cDTW implementation and performance).

Using this approach, we calculated the similarity between time courses for thousands of cells stimulated with six different doses of TGFβ1, grouped them using hierarchical clustering, and defined six response classes of SMAD signaling (Fig EV2D–F, Appendix II.D). The median time courses of the response classes showed qualitatively distinct signaling behavior (Fig 2C). Class 1 is defined by a minimal response to stimulation; its members can therefore be considered non-responders. The other classes show either transient (classes 2 and 3) or sustained dynamics (classes 4–6) of varying levels and duration. As expected, increasing ligand concentrations induced a shift from non-responders toward

transient and then sustained signaling (Fig 2D). However, this transition is not sharp, but gradual, implying that cells from several signaling classes can be observed upon stimulation with a given dose. Accordingly, cells stimulated with the same TGFβ concentration are more distinct in their dynamics than cells grouped into a common signaling class: This was visualized by a higher number of cells with positive silhouette scores in the lower versus the upper panel of Fig 2E. Positive silhouette scores indicate that trajectories were more similar to others in their own group compared to any other group according to cDTW scores (see also Appendix II.D).

We next investigated whether phenotypic responses are primarily determined by the extracellular concentration of the ligand or by the dynamics of SMAD signaling. To this end, we analyzed TGFβ-induced changes in proliferation for all cells belonging to a signaling class or treated with the same extracellular stimulus. We observed that in general, SMAD signaling activity correlated with reduced cell divisions as expected. Sorting cells according to signaling classes indicated that sustained accumulation of SMAD in the nucleus affected cell cycle progression more profoundly then transient SMAD translocation (Fig 2F). Cell motility was altered both by transient and sustained SMAD signaling, although changes remained modest for the first 24 h after (Fig 2G). We detected more robust increases in motility when directed movements were analyzed for a 60-h period post-stimulation (Fig EV2G and H). In all cases, signaling classes provided a better separation of phenotypic outcomes compared to ligand concentration as judged by the magnitude of effects and the appearance of gradual differences between groups (Figs 2F–G and EV2I–J) This supports our hypothesis that the dynamics of signaling, and not the stimulus dose, encode for cellular behavior.

### Dynamics of SMAD signaling are determined by the state of individual cells

Our results so far suggest that heterogeneity in the signaling pathway disturbs transmission of the extracellular signal, that is, the ligand concentration. As a consequence, cells respond to a given input with individual SMAD dynamics that can be grouped in signaling classes. What determines which signaling class a cell belongs to? Previous studies investigating single-cell responses suggest at least three potential sources of cell-to-cell variability: cell cycle, local density, or variations in protein levels (Loewer & Lahav, 2011; Snijder & Pelkmans, 2011).

To determine whether cell cycle state impacts TGFβ signaling, we imaged cells for 24 h before stimulating them with different TGFβ1 concentrations (Fig EV3A). We then sorted cells either according to the last division before the stimulus or according to the amplitude of the response. However, we did not observe any obvious correlation between cell cycle state and SMAD signaling competence (Figs 3A and EV3B). To quantify their relation, we mapped SMAD signaling responses for each individual cell in the new dataset to the previously defined signaling classes (Fig 3B). This mapping was achieved by calculating Euclidian distances of single-cell time courses in both datasets and assigning new trajectories to the signaling class of the most similar single-cell response from the previous experiment (Appendix II.H). As expected, we observed similar distributions of cell division times for all signaling

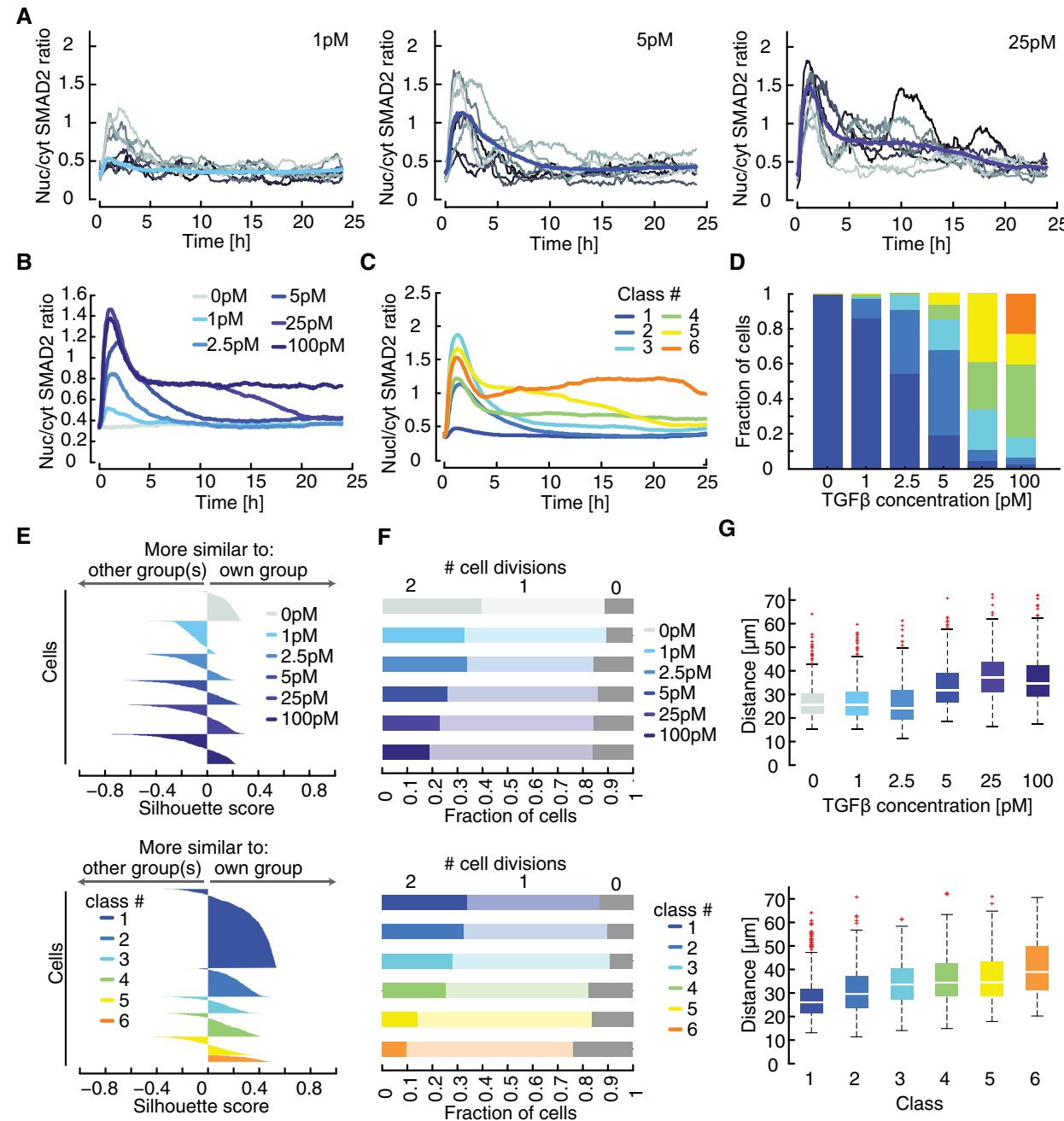

**Figure 2.  SMAD dynamics decompose into distinct signaling classes.**

A   Time-resolved analysis of SMAD2 nuclear to cytoplasmic localization for varying stimulus levels. Nuc/cyt SMAD2 ratios for eight individual cells (thin lines) as well as the population median (thick line) are shown. See Appendix Table S1 for number of cells analyzed.

B   Median nuc/cyt SMAD2 ratio of cells stimulated with varying concentrations of TGFβ1 over 24 h. See Appendix Table S1 for number of cells analyzed.

C   Individual cells were clustered into six signaling classes according to their time-resolved nuc/cyt SMAD2 ratio using dynamic time warping (DTW). Each line represents the median over all cells of the indicated cluster. Cells stimulated with varying TGFβ1 concentrations as indicated in (B) were included in the analysis.

D   Distributions of signaling classes depending on TGFβ dose.

E   Silhouette plots of cells sorted according to TGFβ concentration (upper panel) or signaling classes (lower panel). Plots provide a graphical representation of how well the nuc/cyt SMAD2 ratios of each cell correspond to trajectories of other cells in its own group according to the cDTW measure. Positive silhouette scores indicate that SMAD2 responses are more similar to the own group, while negative scores signify that the corresponding trajectory is closer to any of the other groups. In general, signaling classes provide better separation than sorting according to stimulus levels.

F   Cell proliferation shown as number of cell divisions per cell within 24 h after a TGFβ stimulus. Cells were sorted according to TGFβ concentrations (upper panel) or signaling classes (lower panel).

G   Motility of each cell as summed distance covered between 20 and 24 h after stimulation with TGFβ (in pixel). Cells were sorted according to TGFβ concentrations (upper panel) or signaling classes (lower panel). White lines indicate median; boxes include data between the 25th and 75th percentiles; whiskers extend to maximum values within 1.5× the interquartile range; crosses represent outliers. See Appendix Table S1 for number of cells analyzed.

classes (Figs 3C and EV3C). We further excluded a cell cycle effect using a synchronization protocol: Cells arrested in G2 showed a median TGFβ-induced SMAD2 translocation indistinguishable from an unsynchronized population (Fig EV3D).

As our data indicated that heterogeneity in SMAD2 signaling is independent of cell cycle state, we next investigated whether SMAD signaling of a given cell is influenced by the number and distance of its neighbors. To this end, we calculated a local cell density score for each cell of the population based on the weighted distance of cells in a 640 μm radius (Fig EV3E, Appendix II.E). We found that cell density is not sufficient to explain signaling heterogeneity under our reference culture conditions, as the distribution of density scores was overlapping for all signal classes (Figs 3D and EV3F). Finally, we used the information theoretical measures mutual information and entropy to calculated to which extent signaling heterogeneity can be explained by cell cycle and cell density and determined an

upper bound of below 5% for each process (Fig EV3G and H, Appendix II.E).

Having excluded a major role for cell cycle and cell density, we asked more generally whether signaling heterogeneity arises from differences in the cellular state or from stochastic dynamics of the signaling pathway itself. Previous work on other signaling pathways had shown that sister cells analyses can help tackling this question (Geva-Zatorsky *et al*, 2006; Spencer *et al*, 2009; Sandler *et al*, 2015): If recently divided cells show similar signaling responses, heterogeneity likely arises from cellular state which is assumed to be similar for both sister cells. In contrast, a divergent response in sister cells would indicate that the signaling response is intrinsically unpredictable and stochastic. To analyze the response of sister cells upon TGFβ stimulation, we used a dataset of over 6,000 cells from 11 independent experiments, all treated with 100 pM TGFβ1, and identified cell division events at any time point after stimulation

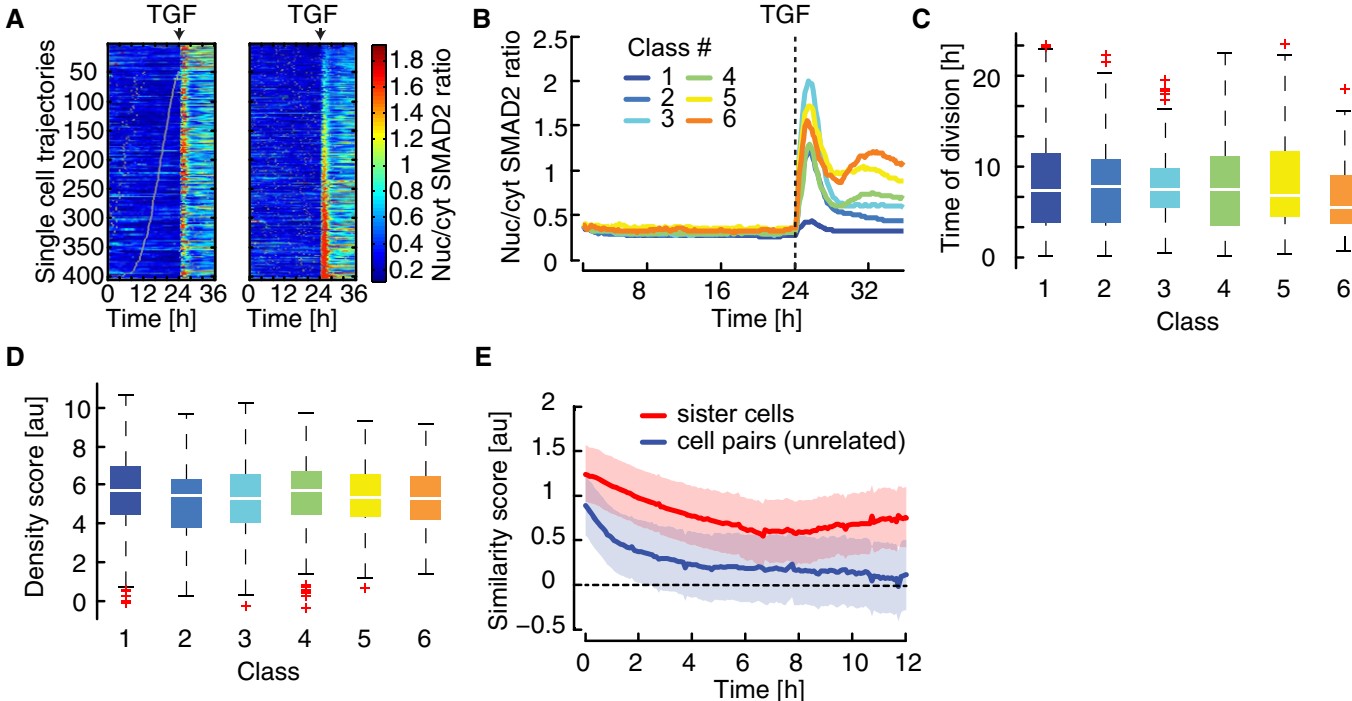

**Figure 3. Heterogeneity in SMAD dynamics determined by cellular state.**

A   Heat map of SMAD2 translocation in individual cells over time. Cells were imaged for 24 h before stimulation with 100 pM TGFβ1. Each horizontal line represents a single cell, and the nuc/cyt ratio is shown as indicated in the legend. Time of cell division is indicated by white marks. Cells were sorted either by the time of the last division before stimulation (left) or by the amplitude of their response (right). Cell cycle and response are not correlated. See Appendix Table S1 for number of cells analyzed.

B   Mapping of SMAD2 translocation dynamics in individual cells to previously identified signaling classes (compare Fig 2C). Cells were imaged for 24 h before stimulation with varying TGFβ1 concentrations (Fig EV3A). For each trajectory, the most similar signaling class was determined using Euclidian distance to the median dynamics of the previously defined clusters (Fig 2C) as a similarity measure. Median nuc/cyt SMAD2 ratios for resulting mapped subpopulations are shown. See Appendix Table S1 for number of cells analyzed.

C   Time of last cell division before stimulus for each signaling class (defined in B). Distributions are overlapping; no significant trend in cell division time is observable. White lines indicate median; boxes include data between the 25th and 75th percentiles; whiskers extend to maximum values within 1.5× the interquartile range; crosses represent outliers. See Appendix Table S1 for number of cells analyzed.

D   Cell density before stimulus for each signaling class (defined in B). Density scores represent a weighted sum of all neighboring cells within 640 μm distance. Distributions are overlapping; no significant trend in cell density is observable. White lines indicate median; boxes include data between the 25th and 75th percentiles; whiskers extend to maximum values within 1.5× the interquartile range; crosses represent outliers. See Appendix Table S1 for number of cells analyzed.

E   Analysis of SMAD2 translocation dynamics in sister cells. SMAD2 translocation dynamics in sister cells after division and unrelated cell pairs with the same nuc/cyt SMAD2 ratio were compared using cDTW. Resulting similarity scores were aligned in time and compared to those from randomly selected cell pairs. Effect size (solid lines) and 95% confidence intervals (shaded areas) were estimated by bootstrapping. The analysis shows that recently divided cells are more similar than control cell pairs and remain correlated over time, indicating that heterogeneity arises from differences in cellular state. See Appendix Table S1 for number of cells analyzed.

(Appendix II.G). We then tracked both sister cells and quantified their divergence by calculating cDTW distances of the corresponding SMAD2 time courses in a sliding window after division events. The cDTW distances were aligned to the time of sister cell division, and compared to a set of control cells that by chance had identical nuc/cyt ratios at a certain time point. Both sister cell and control groups were normalized to the average cDTW distance of random cell pairs (Appendix II.G). Upon alignment of division times, this approach yields an estimation of the time-dependent divergence of sister cells. Interestingly, we found the SMAD2 response to be more correlated in sister cells than in control cell pairs (Fig 3E). While correlation between control cell pairs was lost rapidly within 4 h after stimulation, similarity between sister cells decreased more slowly and sister cells remained significantly correlated throughout the observation period. This indicates that cellular state is an important source of variability, and that the signaling pathway itself responds to a large degree deterministically.

While sister cells showed a correlated response after division, their similarity decreased with time and reached a minimum at around 6 h (Fig 3E). Similar signaling divergence times were reported in previous sister cell studies, where heterogeneity had been attributed to stochastic expression of signaling proteins (Geva-Zatorsky *et al*, 2006; Spencer *et al*, 2009). We therefore hypothesized that SMAD2 signaling heterogeneity is mainly caused by varying concentrations of signaling proteins. While this hypothesis is difficult to test experimentally, it predicts that a deterministic ordinary differential equation model of the signaling pathway would be able to reproduce the population heterogeneity if protein concentrations are sampled from biologically relevant distributions (log-normal distributions; Newman *et al*, 2006).

### A mechanistic model describes population-average SMAD signaling dynamics

To test this prediction, we devised a three-tiered modeling strategy (Fig 4A): We initially derived a mechanistic model of the signaling pathway based on previous literature and calibrated it to median responses of cell populations. Advancing in resolution, we then derived six subpopulation models by fitting to the median time courses of the observed signaling classes, allowing only variation in the expression of signaling proteins and leaving kinetic parameters fixed to their population-average value. Finally, we generated populations of single-cell models by repeated simulation of each subpopulation model with sampling of signaling protein concentrations from log-normal distributions. The final cell population was assembled by combining single-cell simulations from all subpopulations according to the proportions of signaling classes observed in the experimental data.

The topology of the SMAD signaling model comprises three main modules (Fig 4B, Appendix III.A and III.B): The receptor module describes receptor–ligand binding and trafficking of TGFβ receptors between plasma membrane and endosomal compartments (Di Guglielmo *et al*, 2003; Zi *et al*, 2011; Vizan *et al*, 2013). The SMAD module includes receptor-mediated phosphorylation of SMAD2, complex formation with SMAD4, nucleo-cytoplasmic shuttling of SMAD complexes and signal termination by nuclear dephosphorylation of SMAD2 (Schmierer & Hill, 2005; Schmierer *et al*, 2008; Vizan *et al*, 2013). The feedback module describes SMAD-induced

expression of a generic feedback regulator, which acts by inhibiting TGFβ receptors. It represents the combined activity of inhibitory molecules such as SMAD6, SMAD7, and SMURFs (Chen & Meng, 2004; Legewie *et al*, 2008).

The kinetic parameters of the mass action-based ordinary differential equations (ODEs) were estimated by simultaneously fitting the model to time-resolved population-average data of nuclear SMAD2-YFP and SMAD4-YFP translocation for varying doses of TGFβ1 (Figs 4C and EV4A). To further constrain the receptor and feedback modules, we fitted time-resolved Western blot data of receptor levels as well as perturbation experiments in which TGFβ1 was repeatedly added to the medium or receptor signaling was halted using the TGFβ receptor inhibitor SB-431542 (Fig EV4B–G and Appendix Table S4 and Appendix III.C and III.D). The resulting best-fit model represents the average behavior of the cell population and explained the *N* fitted data points within experimental variation ($\chi^2 = 5019; N = 4{,}992$).

We next asked whether our mechanistic model can correctly predict the dynamics of SMAD signaling for previously untested experimentally conditions. To assess the robustness of our model predictions, we analyzed 30 independent model fits with a similar goodness of fit obtained from local multistart optimization (see Appendix III.D). Notably, only few kinetic parameter values in the model could be identified based on the available data and were confined to narrow ranges in all 30 fits (Table EV1, Appendix Fig S4 and Appendix III.E). Nevertheless, all models robustly predicted that signaling is terminated once TGFβ in the medium is depleted by cellular uptake and lysosomal degradation (Massagué & Kelly, 1986; Koli & Arteaga, 1997; Clarke *et al*, 2009). To test this, we measured extracellular TGFβ concentration using a luciferase-based reporter system (Abe *et al*, 1994) and found that ligand decay at an initial TGFβ1 concentration of 25 pM was completed within 20 h as predicted (Fig 4D), coinciding with SMAD2 exit from the nucleus (Fig 4C). We further characterized signal termination by restimulating cells at different time points after an initial 5 pM stimulus. As predicted by the models, only restimulation at a late time point led to a notable response, indicating that the pathway shows refractory behavior early after an initial TGFβ input (Fig 4E and F). This refractory period is prolonged upon strong stimulation, as the SMAD response was unaffected by adding additional ligand at all time points after an initial 100 pM stimulus (Figs 4G and EV4H). Finally, we pre-incubated MCF10A cells with the general transcription inhibitor DRB 30 min before TGFβ1 stimulation to test the model prediction that transcriptional negative feedback shapes the dynamics of SMAD signaling. In line with model predictions, we found that DRB increases the signaling amplitude after stimulation with 100 pM TGFβ1 both at peak time and during later signaling phases (Fig 4H). Taken together, these results show that our deterministic model faithfully reflects the average dynamics of SMAD signaling in populations of cells. Model predictions were robust despite limited parameter identifiability as they most likely depend on identifiable combinations of parameters.

### Varying protein levels determine heterogeneous SMAD signaling

Having implemented a plausible population-average model of the SMAD pathway, we next set out to test if variation in the concentration of signaling proteins is sufficient to explain the observed

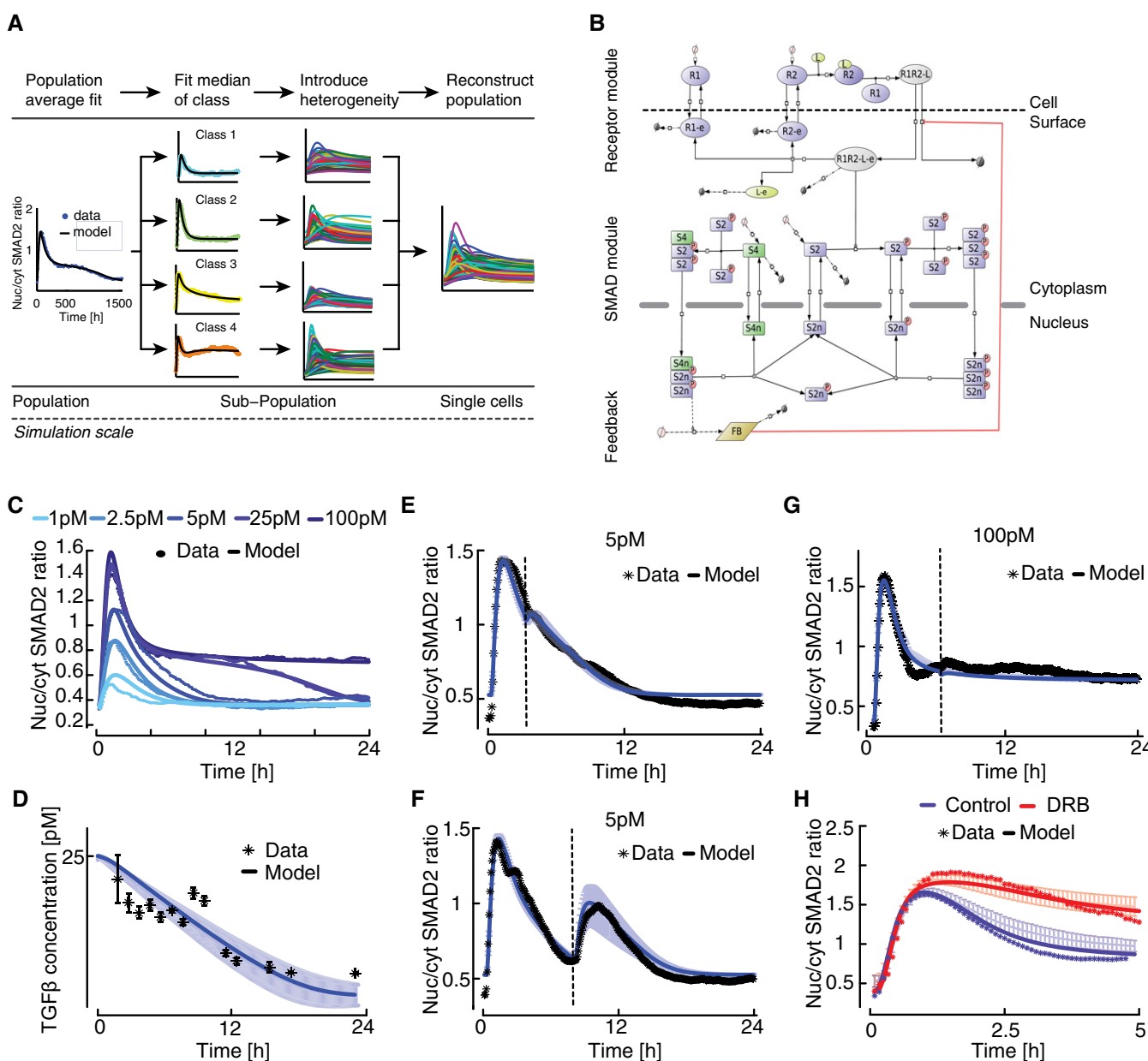

**Figure 4. Mathematical modeling of TGFβ signaling.**

A    Outline of a tiered approach to model heterogeneous signaling in single cells (see text for details).

B    Topology of TGFβ pathway model. The oval shapes represent free receptors (blue), ligand (yellow), and ligand–receptor complex (gray). Extension "-e" signifies endosomal species. Rectangles represent SMAD2 (blue), SMAD4 (green), and generic feedback regulator (yellow). Extensions "p" indicate phosphorylated and "n" nuclear species. Production and degradation are shown by phi symbols. State transitions and intercompartmental shuttling are indicated with arrows, enzyme catalysis with circle headed bars, and feedback inhibition with blunt headed bars.

C    Calibration of population-average model by fitting to median SMAD2 translocation dynamics of cells stimulated with different TGFβ concentrations. Experimental data points correspond to Fig 2B. Model fits to other datasets are shown in Fig EV4 (see also Appendix Table S4); parameter values are provided in Appendix Table S5 and Table EV1.

D    Medium TGFβ degradation over time. Blue line shows the ligand concentration after an initial stimulus with 25 pM TGFβ1 as predicted by the best-fit mathematical model. Shaded area represents the range of predictions from 30 fits with similar goodness of fit obtained from local multistart optimization (see Appendix III.D). Black stars indicate corresponding experimental measurements. Error bars represent standard deviation from three replicates.

E–G  Time-dependent restimulation of the TGFβ pathway at varying input levels. Measured median nuc/cyt SMAD2 ratios (*) and model predictions (—) are shown. Solid lines represent the best-fit model and shaded areas the range of predictions from 30 independent fits (see D). Dashed vertical lines indicate time of second stimulus, which replenishes the extracellular ligand pool to its initial concentration. (E) 5 pM TGFβ1 was applied at 0 h and 3 h. (F) 5 pM TGFβ was applied at 0 h and 8 h. (G) 100 pM TGFβ1 was applied at 0 h and 8 h. See Appendix Table S1 for number of cells analyzed.

H    Effect of the global transcriptional inhibitor DRB on SMAD signaling. Cells were stimulated with 100 pM TGFβ1 in the presence or absence of DRB. Measured median nuc/cyt SMAD2 ratios (*) and model predictions (—) are shown. Solid line represents the best-fit model and shaded area the range of predictions from 30 independent fits (see D). See Appendix Table S1 for number of cells analyzed.

cell-to-cell variability and decomposition into signaling classes. To this end, we quantitatively described signaling classes upon stimulation with 100 pM TGFβ1 by fitting six subpopulation models to the average cluster dynamics (Fig 5A, $\chi^2$ = 1957.8; $N$ = 1,723). These subpopulation models comprised the same kinetic parameter values as the population-average fit, only signaling protein concentrations (e.g., TGFβ receptors or SMAD transcription factors) were allowed to change within a range of 0.5- to twofold around the initial value corresponding to the typical cell-to-cell variation observed for intracellular proteins (Appendix IV.A; Sigal et al, 2006a; Feinerman et al, 2008; Spencer et al, 2009).

Finally, we converted subpopulation models to an ensemble of artificial cells representing the heterogeneity of the entire cell population. Artificial single cells belonging to a signaling class were generated by repeated simulation with signaling protein concentrations varying around the best-fit values of the corresponding subpopulation model (Appendix IV.B). The full cell population was assembled in silico by selecting artificial cells from ensembles according to the proportion of corresponding signaling class observed experimentally at a stimulus of 100 pM TGFβ1 (Fig 2D).

The unknown degree of signaling protein level variation between individual cells was estimated by comparing the SMAD dynamics in simulated populations with experimental measurements from live-cell imaging. To do so, we extracted four signaling features from the single-cell time courses of SMAD2 translocation (100 pM TGFβ1, Fig 5B): the amplitude of the response at about 60 min (E); the plateau after the initial response at about 300 min (L); the ratio of these two quantities characterizing the degree of signal adaptation (E/L); and the time of the maximal nuc/cyt SMAD2 ratio (T). The distribution of these features among the cell population was assessed and the deviation of simulated and measured distributions quantified as a sum of squared distances (Fig 5C, Appendix IV.B and IV.C). This model-data comparison was done while assuming that protein level variation consists of a linear combination of two log-normally distributed noise contributions: a correlated noise that simultaneously affects all signaling proteins in a given cell simulation and arises from fluctuations in the global gene expression machinery (e.g., RNA polymerases and ribosomes), and an uncorrelated noise specific for each signaling protein arising from stochasticity in gene expression (Elowitz et al, 2002; Sigal et al, 2006b; Rhee et al, 2014; Sherman et al, 2015). For simplicity, we assumed the same extent and type of variation for all signaling proteins. By systematically altering the magnitude of correlated and uncorrelated fluctuations, we observed that simulated cell populations robustly matched the experimental measurements over a wide range of noise levels around an optimal combination of both values (Fig 5C). Importantly, using these noise levels, the heterogeneity of the same signaling features at a lower TGFβ concentration could be successfully predicted without further fitting (Fig 5E). The total signaling protein concentrations in the assembled population were continuous and log-normally distributed as expected for biological cell populations (Fig EV5A).

To assess whether our tiered modeling approach with quantitative fitting of signaling classes improves the description of cellular heterogeneity, we compared our results to a simpler modeling approach in which signaling protein concentrations were directly sampled around the best-fit values of the population-average model (Spencer et al, 2009; Paulsen et al, 2011; Gaudet et al, 2012).

Interestingly, this simpler ensemble model described the experimental data less well and was more sensitive to variation in the correlated and uncorrelated noise contributions (Fig 5D and Appendix IV.C). Taken together, our modeling approach indicates that variation in signaling protein concentration is sufficient to quantitatively explain cell-specific SMAD dynamics.

## Negative feedback determines cell-specific responses to TGFβ

Having single-cell simulations reflecting cellular heterogeneity at different TGFβ concentrations at hand, we asked whether our model gives rise to the same proportions of signaling classes as experimentally observed. Therefore, we mapped simulated SMAD2 trajectories from the artificial cell population to the previously observed signaling classes, which resulted in distributions consistent with the experimental data (compare Figs 2D and 6A). Importantly, as for the measured data, grouping simulated cells according to signaling classes yielded a more homogenous separation than grouping according to stimulus strength (Fig EV5B).

Using these simulations, we further investigated features of cellular heterogeneity that are not directly accessible experimentally, and analyzed how cells transition between signaling classes with increasing TGFβ stimulus (Fig 6B). Interestingly, we observe a massive transition from non-responding and transient signaling (classes #1–3) to sustained pathway activation (classes #4–6) between 5 and 25 pM TGFβ1. Model analysis indicates that the switch to sustained signaling emerges because external TGFβ rapidly decays within ~10 h for 5 pM TGFβ1, whereas it remains elevated for about 20 h at 25 pM (Fig 4D). Yet, subpopulation of cells with transient signaling persist at 25 and 100 pM (classes #1–3), indicating that SMAD signaling is restricted despite the continuing presence of ligand, possibly due to high activity of transcriptional negative feedback. To confirm this hypothesis, we systematically lowered feedback expression in artificial cells and observed a strong accumulation of cells with high intensity signaling as expected (Fig 6C; class #6). Importantly, cells with none or transient SMAD activation (classes #1–3) completely disappear upon depletion of feedback in the model, providing evidence that signal termination in these subpopulations indeed relies on negative feedback. Similar results were obtained upon stimulation with 25 pM TGFβ1, while transient signaling classes persisted at 5 pM TGFβ1 even in the absence of feedback (Fig EV5C). Importantly, these model predictions were robust despite uncertainties in the estimated kinetic parameter values (Fig EV5D, Table EV1 and Appendix IV.D). Thus, feedback regulation may underlie the decomposition into qualitatively distinct signaling classes at high TGFβ concentrations.

To further confirm the role of feedback in decomposing SMAD responses into signaling classes, we analyzed signaling protein distributions for each of the six signaling classes using independent model fits with a comparable goodness of fit (Appendix Fig S5A). As these distributions were complex without any parameter providing a clear discrimination between signaling classes, we employed methods from information theory and determined the entropy of model parameters in our subpopulation models (Fig 5A; Appendix IV.D). If the fitted protein concentration values are similar in all subpopulation models, they contain little information to distinguish between response classes and its entropy will be close to maximum (2.6 bits). The more heterogeneous parameter values are

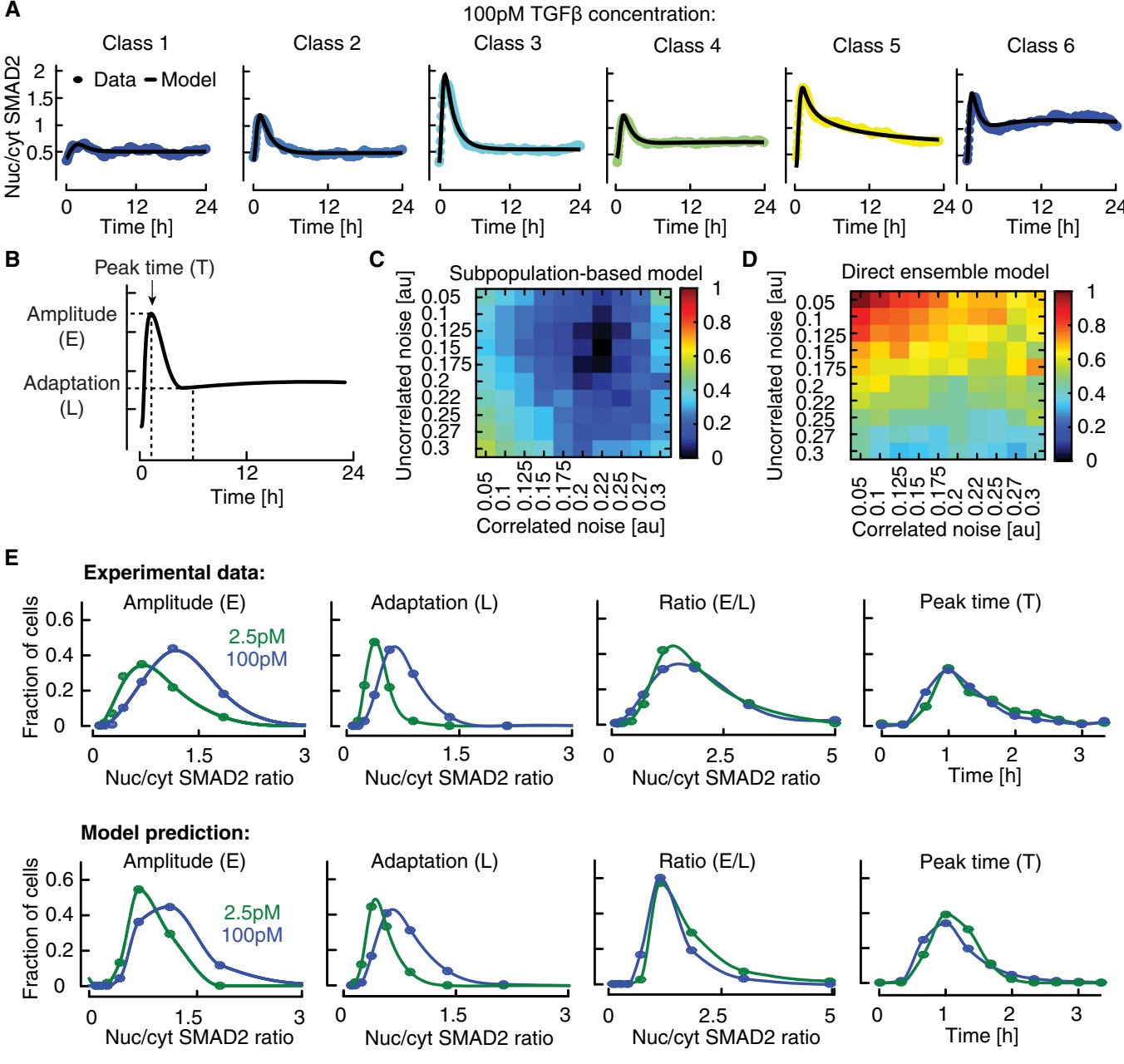

**Figure 5.  Modeling heterogeneous signaling dynamics in single cells.**

A       The model of TGFβ signaling was fitted to six signaling classes observed upon stimulation with 100 pM TGFβ1. Median nuc/cyt SMAD2 ratios (circles) and model fits (solid lines) are shown.

B       Features of SMAD2 translocation dynamics. We considered the amplitude (E) and timing (T) of the first peak of nuclear translocation as well as the amplitude at 300 min (L) as a measure for the signaling activity upon adaptation of the pathway.

C, D   Model performance at varying noise levels. Heterogeneous signaling in response to a 100 pM TGFβ1 stimulus was simulated by signaling class-based modeling (C) or a direct ensemble modeling (D) (see main text). Noise in protein expression is modeled as a combination of correlated and uncorrelated noise (see Appendix IV.B). The differences among single-cell signaling features between model and data are calculated as sum of squared errors and normalized to the maximal deviation observed (color bar). For each combination of correlated and uncorrelated noise, 10,000 cells were simulated.

E       Measured and predicted distributions of signaling features for two TGFβ stimuli (2.5 and 100 pM). A population of artificial cells was assembled according to signaling class distributions observed upon stimulation with 100 pM TGFβ1 using optimal noise contributions (see panel C). Signaling features were extracted from simulations at different TGFβ concentrations.

among subpopulations, the lower the measured entropy is and the more they may contribute to the divergent signaling dynamics of the classes (Fig 6D). While many signaling protein concentrations show

relatively similar values in all subpopulation models (entropy ~2.6 bits), the level of feedback protein indeed carried the most information to distinguish between signaling classes.

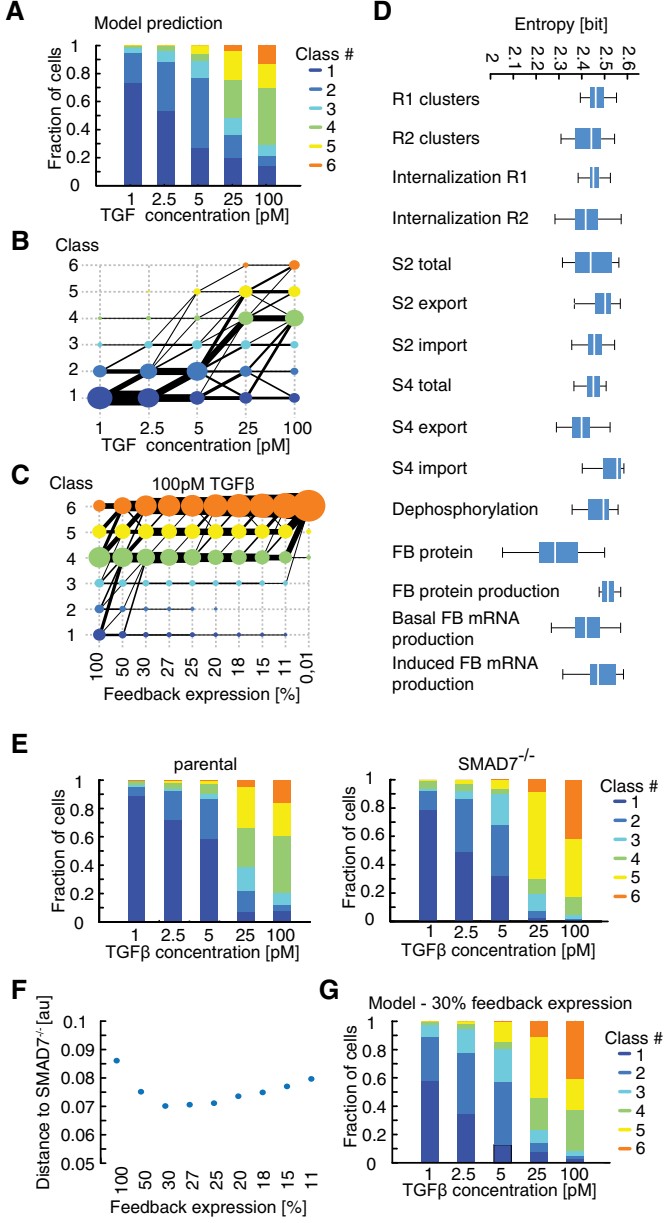

Figure 6.  Negative feedback determines decomposition into signaling classes.

A  Predicted distributions of signaling classes depending on TGFβ dose. Simulations were performed as described for Fig 5E. The simulated time courses were mapped onto the original clusters dynamics (Fig 2C) as described in Appendix II.H.

B  Transition between signaling classes depending on stimulus strength. Same data as in (A). Black lines and their thickness indicate the direction and extent of transitions between signaling classes. Filled circle size indicates the proportion of artificial cells in the corresponding signaling class.

C  Transition between signaling classes depending on feedback strength. The response of a reassembled population of artificial cells to 100 pM TGFβ1 was simulated with reduced feedback expression as indicated (see Appendix IV.D) and mapped to previously observed signaling classes. Black lines and their thickness indicate the direction and extent of transitions between signaling classes. Transitions with a probability below 1% were excluded for better visualization.

D  Variation of model parameters across signaling classes. For 30 independent model fits to the experimentally observed signaling classes upon stimulation with 100 pM TGFβ1 (see Appendix), the variation of the indicated parameters between signaling classes was calculated as entropy. Lower entropies indicate more variation between signaling classes; uniform parameter distribution would lead to the maximal entropy of 2.6 bits. White lines indicate median; boxes include data between the 25th and 75th percentiles; whiskers extend to maximum values within 1.5× the interquartile range.

E  Distribution of signaling classes in parental and SMAD7 knock-out cells. Cells were stimulated with indicated concentrations of TGFβ and measured SMAD2 translocation dynamics mapped to the previously observed signaling classes (Fig 2C).

F  Calibration of feedback level. Signaling class distributions at varying levels of feedback expression (C) were compared to experimentally observed distribution upon SMAD7 knock-out (E). Minimal divergence between model and data was observed at 30% feedback expression.

G  Predicted distributions of signaling classes depending on TGFβ dose at 30% feedback expression. Simulations were mapped to the previously observed signaling classes (Fig 2C)

To experimentally test the predicted role of feedback in signaling heterogeneity, we deleted SMAD7 in SMAD2 reporter cells using Cas9-mediated gene knock-out (Fig EV5E and F). SMAD7 is considered to be one of the main feedback regulators of TGFβ-induced signaling, and acts at the level of TGFβ receptors as implemented in our model (Moustakas & Heldin, 2009). We measured SMAD2 dynamics in response to various doses of TGFβ1 in the parental and knock-out cell line and mapped the resulting time series to the initial observed signaling classes (Figs 6E and EV5G). As predicted by the model, we observed a shift in signaling classes toward those with higher signaling strength. We next aimed to compare the measured single-cell responses to model simulations. As we assumed that some feedback activity is retained in SMAD7 knock-out cells due to the presence of redundant transcriptional feedback regulators in TGFβ signaling (Wegner et al, 2012), we compared signaling class distributions from experimental data and model simulations with

varying feedback strength and observed the best match at 30% feedback strength (Fig 6F and G and Appendix IV.D). In both model and experiment, feedback depletion led to a disappearance of the non-responding and transient classes #1–3 at high doses of TGFβ1 (25 and 100 pM). In contrast, cells remained in transient signaling classes at or below 5 pM TGFβ1, confirming that ligand depletion dominates signal termination at low input levels. Interestingly, loss of SMAD7 did not alter the population-median signal duration (Fig EV5G), further suggesting that it affected this feature only in a subpopulation of cells.

A noticeable difference between model and experiment was that the model predicted a lower fraction of non-responding cells in SMAD7 knock-out cells at TGFβ concentrations below 5 pM when compared to experimental measurements. To explain this discrepancy, we further analyzed parameter variations between signaling classes using independent model fits. We observed that the non-responding signaling class #1 differs from the remaining signaling classes, as it is characterized by a high ratio of feedback protein to receptor levels (Appendix Figs S5B and S6). We hypothesized that knock-out cells compensate the loss of SMAD7 by downregulating TGFβ receptor levels, thereby increasing the feedback-to-receptor ratio and the fraction of non-responding cells. Western Blot analyses support this hypothesis as we observed reduced TGFβRII levels in SMAD7 knock-out cells (Appendix Fig S7). Taken together, we conclude that negative feedback leads to decomposition into

qualitatively distinct signaling classes upon strong stimulation, while its loss can be partially compensated at lower input levels.

In summary, our combined experimental and computational study shows that the response to a given dose of TGFβ1 is determined cell specifically by the levels of certain signaling proteins. This leads to decomposition of cell populations into classes of SMAD2 dynamics, which determine the phenotypic response to a stimulus. Changing the level of negative feedback regulators such as SMAD7 allows shifting the response of a given cell and therefore enables fine-tuned control of the cellular response to TGFβ1 in populations of cells.

## Discussion

Efficient information processing by the TGFβ signaling pathway is crucial during development, tissue homeostasis, and regeneration, as compromised signaling contributes to severe human diseases such as cancer. To predict cellular responses to this versatile cytokine and modulate them by targeted therapies, we need a quantitative understanding of how cells encode and decode information about the strength and duration of the extracellular input. Using a combination of time-resolved measurements at the single-cell level and quantitative mathematical modeling, we reveal that cell-specific long-term dynamics of SMAD nuclear translocation determine the phenotypic response of epithelial cells to TGFβ.

Our experimental approach allowed us to measure the nuclear to cytoplasmic translocation of SMAD2 and SMAD4 with unprecedented time resolution and precision at the single-cell level for up to 60 h. We observed transient SMAD accumulation in the nucleus during the first four hours that, depending on the input strength, was followed by a second signaling phase with temporally less defined periods of nuclear translocation. The average response of our single-cell measurements corresponded well with biochemical measurements in previous studies (Inman *et al*, 2002; Clarke *et al*, 2009; Zi *et al*, 2011; Vizan *et al*, 2013). However, our results conflict with a previous study in single cells that reported transient SMAD4 but sustained SMAD2 nuclear accumulation upon TGFβ stimuli using fluorescent protein-based reporter in mouse myoblast C2C12 cells (Warmflash *et al*, 2012). In addition to cell-type differences, a noticeable distinction in the experimental setup that may explain the contrasting results is the higher level of overexpression of tagged SMAD2 in the previous study (> 2× vs. 0.5× compared to endogenous levels). Moreover, we carefully validated results from live-cell reporters using Western blot and immunofluorescence analysis of endogenous SMAD proteins to exclude perturbations of the signaling network by transgene expression.

Several molecular mechanisms contributed to regulating SMAD dynamics. For the mammary epithelial cell line and culture conditions used in this study, overall duration of pathway activation at low TGFβ concentrations was mainly controlled by ligand degradation due to endocytosis. At higher input levels, initial nuclear accumulation was limited by a combination of ligand degradation, receptor internalization and the activity of transcriptional feedbacks. However, the role of feedbacks at the population-average level was limited as we observed attenuation of nuclear SMAD accumulation even in the presence of the transcriptional inhibitor DRB, consistent

with previous studies using translation inhibitors (Pierreux *et al*, 2000; Inman *et al*, 2002). Moreover, persistent signal attenuation in SMAD7 knock-out cells demonstrated redundancies between transcriptional feedbacks that need to be investigated further. During later signaling phases, periods of SMAD nuclear accumulation were asynchronous and of variable length. While our current understand of the pathway topology does not provide an intuitive understanding of such spontaneous pathway activation, an intriguing hypothesis would be that vesicle-mediated recycling of receptors to the cell surface leads to stochastic increases in the cellular sensitivity to the ligand, as similar processes have been observed in the context of EGF signaling (Villaseñor *et al*, 2015). In further studies, combined live-cell reporters for SMAD translocation and receptor localization may provide deeper insights regarding the molecular mechanisms of sustained pathway activation.

To analyze SMAD translocation dynamics in thousands of genetically identical reporter cells, we established constrained dynamic time warping as a tool for non-linear alignment of time series data. Dynamic time warping both emphasizes similarities in dynamic patterns of the time courses, and allows quantifying differences in signal amplitude, thereby improving the grouping of noisy single-cell trajectories. By allowing for stretching and squeezing of time courses, DTW is less sensitive to asynchronies than simpler similarity measures such as the Euclidian distance. However, constraining the extent of temporal alignments in DTW is critical to ensure that results remain biologically significant. Using DTW-based time course clustering, we observed pronounced cell-to-cell variability at all stimulus levels. Heterogeneous cell responses led to a decomposition of TGFβ signaling into signaling classes with the fraction of cells in each class depending on the stimulus strength. Interestingly, a recent study proposed that the response of MCF10A cells to extracellular ATP can be similarly group in three classes corresponding to distinct cellular states (Yao *et al*, 2016). Although signaling classes represent mathematically identifiable clusters of time courses and provide a more homogenous grouping compared to other determinants such as ligand concentration, it is important to note that SMAD dynamics in each class vary gradually and represent a continuum of response profiles. The definition of six classes therefore remains a heuristic choice to classify the observed heterogeneity. In future studies, it may be interesting to use other approaches established in the context of single-cell sequencing such as diffusion maps to better recover low-dimensional structures underlying our high-dimensional observations (Haghverdi *et al*, 2016).

Many processes have been reported to influence cellular heterogeneity (Loewer & Lahav, 2011; Snijder & Pelkmans, 2011). We found that cell cycle state and cell density provide only minor contributions to variability in the SMAD response of individual cells. For cell density, this is in accordance with a recent publication demonstrating that activation of the cell density sensing YAP/TAZ pathway does not attenuate SMAD signaling (Nallet-Staub *et al*, 2015). Only in polarized cells, apical access is restricted for TGFβ receptors, which may lead to reduced ligand exposure depending on the delivery mode of the stimulus.

To test whether protein level variations may cause signaling heterogeneity and decomposition into signaling classes, we developed a tiered modeling approach based on deterministic subpopulation models fitted to experimentally observed time courses. Our approach is similar to previous work in which heterogeneous

ensembles of single cells were simulated by sampling the signaling protein concentrations around the population median (Spencer *et al*, 2009; Paulsen *et al*, 2011; Gaudet *et al*, 2012). However, the detailed description of defined subpopulations ensured a robust and more precise description of heterogeneity, while minimizing computational cost compared to individually fitting parameters for each cell (Kallenberger *et al*, 2014; Yao *et al*, 2016). It would therefore be easy to translate the concept to other cellular systems where time-resolved data at the single-cell level is available, such as NF-kB or p53 signaling (Nelson *et al*, 2004; Geva-Zatorsky *et al*, 2006; Tay *et al*, 2010). However, the current approach relies on temporally stable differences in protein production rates. While this assumption holds true on limited timescales, it will break down when considering response times longer than cell cycle length. Time-varying production rates may solve this issue but will complicate fitting procedures. Moreover, truly stochastic processes such as the proposed stochastic changes in TGFβ sensitivity during later signaling phases will not be accessible by this approach and require fully stochastic models to provide further insights.

While our modeling approach highlights the importance of protein level variations, the source of these variations remains elusive. Through many studies in the past years, it became evident that protein level variations represent a combination of fluctuations caused by the stochastic nature of biochemical reactions (Bar-Even *et al*, 2006; Pedraza & Paulsson, 2008; Lestas *et al*, 2010), cell-specific activity of regulatory processes (Colman-Lerner *et al*, 2005) and influences from population microenvironment (Snijder *et al*, 2009; Snijder & Pelkmans, 2011). These processes affect mammalian signaling systems to varying degrees (Feinerman *et al*, 2008; Snijder *et al*, 2009; Spencer *et al*, 2009; Kallenberger *et al*, 2014; Frechin *et al*, 2015; Adamson *et al*, 2016). Depending on the lifetime of the associated biomolecules, fluctuations from stochastic processes are supposed to vary on shorter time scales compared to regulated sources of cellular heterogeneity. Our sister cell analysis indicates a fast decaying component (within 6 h) as well as stable differences between cells that last beyond the observation period. As the grouping of cells in signaling classes is relatively stable over time, we assume that the long-lasting component dominates cellular heterogeneity. This may reflect differences in signaling history of individual cells, leading to varying states of the TGFβ network due to the activity of interacting signal pathways (Guo & Wang, 2009). Depending on the strength of the input, these varying states of the pathway will translate into transient or sustained activation of SMAD signaling and therefore a transition of cells between signaling classes. We find that the levels of few signaling proteins are governing these transitions and provide evidence that feedback expression is a main determinant of signaling classes. At this point, we can only speculate how differences in feedback and specifically SMAD7 expression could arise in genetically identical cells. In addition to stochastic gene expression, cell-specific activation of signaling pathways controlling SMAD7 expression could contribute to the observed cell-to-cell variability. Such pathways may include IFN-γ/Stat1 (Ulloa *et al*, 1999), PKC (Tsunobuchi *et al*, 2004), hepatocyte growth factor (Shukla *et al*, 2009) or mir21 (Li *et al*, 2013). Further experiments are needed to clarify sources of heterogeneous feedback expression.

Feedback is an essential part of most signaling pathways (Legewie *et al*, 2008) and is known to support different features of information transmission depending on network topology and kinetic parameters (Leibler & Barkai, 1997; Yi *et al*, 2000; Rosenfeld *et al*, 2002; Yu *et al*, 2008; Voliotis *et al*, 2014). Our analysis indicates that in the TGFβ network, feedback mainly acts at high input levels to limit sustained pathway activation, thus promoting adaptation as reported for other signaling systems (Yi *et al*, 2000; Ma *et al*, 2009). This could be due to non-linear induction of SMAD7 or a stronger contribution of other parameters such as receptor levels at lower ligand concentrations. In contrast to previous studies (Leibler & Barkai, 1997; Yi *et al*, 2000; Paulsen *et al*, 2011), we do not find that negative feedback reduces signaling variability as measured by SMAD2 translocation, but provide evidence that it promotes heterogeneity by establishing signaling classes with transient dynamics at high TGFβ concentrations. Additionally, feedback modulates the amplitude of the response as indicated by transitions within transient and sustained signaling classes, for example, from class 2 to 3 at 5 pM TGFβ or from class 4 to 6 at higher stimulus levels. As our experimental study was limited to SMAD7, it would now be interesting to investigate the contribution of the remaining negative feedbacks. Do they indeed provide redundancy or do they regulate specific features of information transmission?

Our single-cell analysis shows that cell-specific long-term dynamics of SMAD translocation determine the phenotypic response to TGFβ activation. Interestingly, it seems that migration and proliferation may be controlled by different features of SMAD signaling: migration tended to be affected already by a transient SMAD translocation (class 2–3), whereas anti-proliferative effects seemed to require sustained SMAD signaling (class 4, 5, and mainly 6). These findings are consistent with previous studies in cancer cell lines in which transient SMAD activation was sufficient to alter cellular motility and induce EMT–like processes, while sustained signaling was required to influence proliferation (Nicolás & Hill, 2003; Giampieri *et al*, 2009). Hence, our analysis shows that dynamic information encoding observed at the level of cell lines may be conserved at the level of heterogeneous single-cell signaling and reflect the regulatory potential of the pathway: By fine-tuning the level of signaling proteins through interacting signaling pathways, the sensitivity of individual cells to TGFβ inputs can be adjusted within a tissue. This would allow stratifying the cellular response depending on the state of the cell. During therapy, this property of the TGFβ pathway could be exploited by specifically modulating the levels or enzymatic activities of selected proteins to switch the response from EMT-like processes to proliferation control. As TGFβ activity is often tightly linked to tumor progression, such a targeted approach may help to improve therapies against advanced cancers.

## Materials and Methods

### Cell line and constructs

Human breast epithelial MCF10A cells were cultured in DMEM/F-12 medium supplemented with 5% horse serum, 20 ng/ml epidermal growth factor (EGF), 0.5 μg/ml hydrocortisone, 100 ng/ml cholera toxin, and 10 μg/ml insulin, penicillin, and streptomycin (Debnath *et al*, 2003). When required, the medium was supplemented with selective antibiotics to maintain transgene expression (400 μg/ml G418, 50 μg/ml hygromycin or 0.5 μg/ml puromycin). We

generated lentiviral reporter constructs for SMAD2 and 4 using the MultiSite Gateway recombination system (Thermo Fisher Scientific) by fusing the protein coding sequence to the yellow fluorescent protein Venus (YFP) under the control of a constitutive human Ubiquitin C promoter (UbCp). We infected MCF10A cells with corresponding lentiviral particles together with viruses expressing histone 2B fused to cyan fluorescent protein (H2B-CFP) under the control of UbCp as a nuclear marker. Subsequently, stable clonal cell lines were established and validated. To knock-out SMAD7, we first infected SMAD2 reporter cells with lentiviruses expressing Cas9 under control of a doxycycline-inducible promoter (Wang *et al*, 2014b). A stable, clonal cell line was further infected with viruses expressing an sgRNA targeting exon 2 of SMAD7 (TCCTTACTCCA GATACCCGA) (Shalem *et al*, 2014) and cultured for 2 weeks in the presence of doxycycline. Finally, we screened clonal cell lines for alterations of the SMAD7 locus by genomic PCR (Thermo Fisher Scientific) and sequencing and selected a line with non-sense mutations in both alleles (Fig EV5E).

## Antibodies and reagents

We used antibodies against total SMAD2 (D43B4, #5339) and pSMAD2 (Ser465/467, 138D4, #3108) from Cell Signaling, SMAD4 (B-8, #sc-7966) and TGFβRII (E-6, #sc-17792) from Santa Cruz, and GAPDH (#G9545) from Sigma-Aldrich. Recombinant human TGFβ 1 was obtained from R&D Systems (#240-B-002) and stored at $-80°C$ in 4 mM HCl, 1 mg/ml bovine serum albumin at 390 nM. DRB (5,6-dichlorobenzimidazole 1-β-D-ribofuranoside) was purchased from Cayman (used at 100 μM), TGFβRI kinase inhibitor VI SB431542 from Calbiochem (used at 10 μM) and CDK1 inhibitor RO 3306 (used at 3 μM) from Axon.

## Time-lapse microscopy

For live-cell time-lapse microscopy, $2 \times 10^5$ cells were plated in 35-mm poly-D-lysine-coated glass bottom plates (MatTek or ibidi) 2 days before experiments. Before starting the experiment, cells were washed twice with $1 \times$ PBS and media was changed to RPMI lacking phenol red and riboflavin supplemented with all growth factors, 5% horse serum and antibiotics. The microscope was surrounded by a custom enclosure to maintain constant temperature (37°C), $CO_2$ concentration (5%), and humidity. Cells were imaged on a Nikon Ti inverted fluorescence microscope with a Hamamatsu Orca R2 or Nikon DS-Qi2 camera and a 20× plan apo objective (NA 0.75) using appropriate filter sets (Venus: 500/20 nm excitation (EX), 515 nm dichroic beam splitter (BS), 535/30 nm emission (EM); CFP: 436/20 nm EM, 455 nm BS, 480/40 nm EX). Images were acquired every 5 min for the duration of the experiment using Nikon Elements software. TGFβ 1 was prepared in 500 μl media and added, if not noticed otherwise, after one round of images to achieve the final concentration in 2.5 ml media.

## Image analysis and cell tracking

Cells were tracked throughout the duration of the experiment using custom-written MATLAB (MathWorks) scripts based on code developed by the Alon laboratory (Cohen *et al*, 2008) and the CellProfiler project (Carpenter *et al*, 2006). In brief, we applied flat field

correction and background subtraction to raw images before segmenting individual nuclei from nuclear marker images using thresholding and seeded watershed algorithms. Segmented cells were then assigned to corresponding cells in following images using a greedy match algorithm. Only cells tracked from the first to last time point were considered. For most analyses, we tracked cells in forward direction from the first to the last time point. Upon division, we followed the daughter cell closest to the last position of the mother and merged tracks from mothers and offspring. For sister cell analyses, cells were tracked backward from the last to the first time point, tracks from offspring, and mothers were again merged. As a consequence, tracks of sister cells are identical before cell division. We quantified nuclear fluorescence intensity and measured the fluorescence intensity in the cytoplasm using a 4-pixel wide annulus around the nucleus. Finally, we estimated the nuc/cyt ratio for each cell over time and analyzed the resulting single-cell trajectories computationally (Appendix II.A). As nuclear envelope breakdown during mitosis prevented meaningful measurements of SMAD translocation, we interpolated corresponding values. See Appendix for further details on image analysis, cell tracking, and data processing.

## TGFβ measurement

We used Mink lung epithelial cells (MLECs) stably transfected with a reporter containing a truncated PAI-1 promoter (3TP promoter with three consecutive TPA response elements) fused to the firefly luciferase gene and cultured them in 96-well plates using DMEM (Abe *et al*, 1994). Supernatants from live-cell microscopy experiments were removed at defined time points and added in triplicates to MLEC reporter cells. After incubation overnight, cells were lysed and thawed. Luciferase activity was measured by 10-s per well readings on a 96-well format luminometer (see Appendix II.I for details).

## Western blot analysis

Cells were plated 2 days before experiments. After stimulation, we harvested cells at indicated time points and isolated proteins by lysis in the presence of protease and phosphatase inhibitors. Total protein concentrations were measured by BCA assay (Thermo Fisher Scientific). Equal amounts of protein were separated by electrophoreses on 10% SDS–polyacrylamide gels and transferred to PVDF membranes (GE Healthcare) by electroblotting (Bio-Rad). We blocked membranes with 5% non-fat dried milk or 5% bovine serum albumin, incubated them overnight with primary antibody, washed them, incubated them with secondary antibody coupled to peroxidase (#31460, Thermo Fisher Scientific), washed again, and detected protein levels using chemoluminescence (ECL Prime, GE Healthcare). Blots were quantified using ImageJ (Schneider *et al*, 2012).

## Reverse transcription qPCR

Cells were plated 2 days before experiments. Total RNA was extracted using High Pure RNA Isolation kit (Roche), and concentration was determined by using a photospectrometer (NanoDrop 2000, Thermo Fisher Scientific). 1 μg of RNA sample was converted to complementary DNA using M-MuLV reverse transcriptase (NEB)

or Proto Script II reverse transcriptase (NEB) and oligo-dT primers. Quantitative PCR was performed in triplicates using SYBR Green reagent (Roche) on a StepOnePlus PCR machine (Applied Biosystems). Primer sequences: β-actin forward, GGC ACC CAG CAC AAT GAA GAT CAA; β-actin reverse, TAG AAG CAT TTG CGG TGG ACG ATG; SnoN forward, GGCTGAATATGCAGGACAG SnoN reverse, TGA GTT CAT CTT GGA GTT CTT G; SMAD7 forward, ACC CGA TGG ATT TTC TCA AAC C SMAD7 reverse, GCC AGA TAA TTC GTT CCC CCT; PAI1 forward, GGC TGA CTT CAC GAG TCT TTC A; PAI1 reverse ATG CGG GCT GAG ACT ATG ACA.

## Immunofluorescence

Cells were plated 2 days before experiments on coverslips coated with poly-L-lysine (Sigma-Aldrich) and fixed at indicated time points with 2% paraformaldehyde. Cells were permeabilized with 0.1% Triton X-100 in PBS, blocked with 10% goat serum in PBS, incubated with primary antibody in 1% BSA in PBS, washed with 0.1% Triton X-100 in PBS, and incubated with secondary antibody conjugated with Alexa Fluor 488 (#A-11034) or Alexa Fluor 647 (#A-21245, Thermo Fisher Scientific) in 1% BSA in PBS. After washing, cells were stained with 2 μg/ml Hoechst in 0.1% Triton X-100/PBS and embedded in Prolong Antifade (Thermo Fisher Scientific). Images were acquired with a 20× plan apo objective (NA 0.75) using appropriate filter sets. Automated segmentation was performed in MATLAB (MathWorks) with algorithms from CellProfiler (Carpenter *et al*, 2006).

## Computational modeling

Model simulations and fitting were performed using the MATLAB toolbox Data2Dynamics (Raue *et al*, 2015). The implementation of the model and the computational methods are described in Appendix III and IV.

## Data availability

Reporter cell lines are available upon request. The primary datasets and mathematical models generated in this study are available in the following databases:

- Unprocessed single-cell data: Dryad Digital Repository (https://doi.org/10.5061/dryad.hc5dp).
- Mathematical models: BioModels Database (www.ebi.ac.uk/biomodels-main, MODEL1712050001 – MODEL17120500012).
- SED-ML scripts and simulations reproducing Figs 4C–H and 5A: JWS Online Simulation Database (https://jjj.bio.vu.nl/models/experiments/?id = strasen2018).

Expanded View for this article is available online.

## Acknowledgements

We thank K. Janes and J. Brugge for providing MCF10A cells, Gitta Blendinger and Andrea Katzer for excellent technical assistance, and Lennart Schnirch for help establishing the SMAD7 knock-out cell line. SL and US were supported by NIH funding (NIH 1R01DK090347), by the e:bio junior group program (FKZ 0316196) and the Virtual Liver Network (FKZ 0316054) of the German Federal Ministry of Education and Research (BMBF).

## Author contribution

JS and SB performed experiments, MJ data analysis and US mathematical modeling; CS contributed to generating the SMAD7 knock-out cell line; DH and PK performed TGFβ measurements. JS, US, MJ, and AL prepared figures; SL and AL wrote the manuscript with contributions from all authors; SL and AL conceived the study and supervised the research.

## Conflict of interest

The authors declare that they have no conflict of interest.

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
