## [Review Process File · Molecular Systems Biology]

Cell-specific responses to the cytokine TGF β are determined by variability in protein levels

Jette Strasen, Uddipan Sarma, Marcel Jentsch, Stefan Bohn, Caibin Sheng, Daniel Horbelt, Petra Knaus, Stefan Legewie & Alexander Loewer

Review timeline:

Submission date:	6 May 2017
Editorial Decision:	16 June 2017
Revision received:	14 September 2017
Editorial Decision:	30 November 2017
Revision received:	12 December 2017
Accepted:	15 December 2017

Editor: Maria Polychronidou

Transaction Report:

1st Editorial Decision

16 June 2017

Thank you again for submitting your work to Molecular Systems Biology. We have now heard back from two of the three referees who agreed to evaluate your study. Since their recommendations are similar, I prefer to make a decision now rather than delaying the process. As you will see below, the reviewers think that the presented findings are interesting. They raise however a series of concerns, which we would ask you to address in a revision of the manuscript.

The reviewers' recommendations are rather clear so I think that there is no need to repeat the points listed below. Of course, feel free to contact me in case you would like to discuss any particular point in further detail.

REVIEWER REPORTS

Reviewer #1:

Summary:

The authors aim to elucidate the origins and causes of cell-to-cell heterogeneity in TGF- β induced SMAD translocation to the nucleus. To achieve this, they generated clonal cells lines containing YFP tagged SMAD2 and SMAD4 constructs. Using automated tracking and cell segmentation algorithms (based on H2B-CFP) they manage to obtain long term single-cell dynamics of SMAD translocation. They observe a substantial degree of cell-to-cell heterogeneity and can identify six distinct signaling clusters of SMAD responses across all used doses of TGF- β . They argue that a cell's phenotypic response does not depend on the concentration of the exogenous input but on its

specific signaling response. The latter is correlated to, but not exclusively dependent on the concentration of TGF- β . The authors make the point that a cell's signaling response is independent of its cell cycle stage and the extent of local cell crowding it experiences. Nevertheless, they argue that variability in single-cell signaling response is largely deterministic as sister cell responses decorrelate slower than a pair of random cells, indicating that information is encoded in the cellular state. These predetermined differences in cellular state are then implied to be differences in protein concentrations and/or the propensity of expressing feedback mediators of TGF- β signaling. An ODE based model of TGF- β signaling is constructed and parameters estimated. The model suggests that TGF- β degradation, initial protein concentrations and expression levels of feedback mediators (SMAD7) are the main variables shaping the response of SMADs. By using CRISPR/Cas9 mediated knockout of SMAD7 the authors demonstrate that the fraction of sustained responders in a cell population can be increased.

General Remarks:

The study is generally of high quality. The experiments are carefully performed, and the quantifications appear thoroughly done. In general, this is a fine paper, and even though the phenomenon of cell-to-cell variability in signaling responses in mammalian cells has now been studied for the last 13 years, it remains a topic of high relevance. In particular, the authors reveal that most of the variability appears to be of a non-stochastic origin, which is an important message that is still not widely accepted in the community. For this reason, the paper deserves, provided the evidence is strong, to be published in a high-profile journal such as *Molecular Systems Biology*. Perhaps the main weakness of the study (besides the points listed below, some of which are major), is that the mechanistic explanation of what then determines the observed cell-to-cell variability remains rather shallow. My general take on the approach used, namely to identify causes of variability by trying to explain differences between single cells by tweaking a simple ODE model of a few interacting components, is nice but remains very hypothetical unless the identified parameters that are causative are experimentally proven to drive cell-to-cell variability. In this light, their SMAD7 knockout experiments are certainly laudable. My use of the term 'shallow' reflects the fact that there is no explanation of what determines differences in the expression level of SMAD7. But maybe I am asking too much. Perhaps the very least the authors could do is speculate how in growing populations of genetically identical adherent cells that are exposed to identical conditions differences in SMAD7 expression levels could arise and that this is still an open question.

Points of criticism:

Figure S1G shows quite different dynamics and wider distributions for YFP-SMAD2 activity than for endogenous SMAD2. Is YFP-SMAD2 a bona fide reporter, or should the authors have chosen for fluorescent protein tagging of endogenous SMAD, for instance using CRISPR/Cas9? This is a crucial point to be able to derive meaningful conclusions from the analysis of cell-to-cell variability. The authors use a defined, non-adaptive region in the cytoplasm which is solely dependent on the nuclear segmentation to quantify the cytoplasmic signal of YFP-SMAD2 and use the intensity in this region as a proxy for the whole cytoplasmic intensity. This is commonly used for proteins which exert their function by nuclear translocation. However, in figure 1D it seems that the cytoplasmic distribution of SMAD2-YFP is far from homogenous (E.g. 100 and 120 min). Thus, differences in the cytoplasmic/nuclear ratio between individual cells could be an artifact of localization differences in the cytoplasm. More critically, this might change as a function of cell area and shape which would raise major issues with respect of arguing that the cellular phenotype is not causative for the distinct signaling clusters. Is there any possible way of excluding this explanation?

In figure 1B two distinct bands are visible for YFP-SMAD2 and only one can be observed for endogenous SMAD2. Could the authors elaborate on why this is the case? It is of interest since if there are two distinct forms of YFP-SMAD2 present in the population, the observed single-cell signaling response clusters might represent distinct mixtures of these two YFP-SMAD2 isoforms in each cell.

Figure 1C shows dynamics of phosphorylation of SMAD2 and YFP-SMAD2 and the authors argue that the dynamics are comparable. Judging from the blots it appears that pYFP-SMAD2 decreases substantially faster after stimulation than its endogenous counterpart. A quantification would be helpful. Moreover, one sees a double band also for pYFP-SMAD2 but the upper band seems to be less phosphorylated than the lower band. What does that mean?

The authors extensively describe how individual cells are tracked, segmented and signal is quantified. However, there is no description of how mitotic cells are treated with respect to assigning them to the signaling classes. This is especially relevant in figure 1F. E.g. when a cell

divides at five hours after stimulation how can a distinction between classes two and four be made which look remarkably similar until that point. Are the offspring's dynamics included and merged with its mother's dynamics?

Figure 2F tries to exemplify that the fraction of dividing cells, i.e. in their argument the phenotypic response of individual cells, is determined by a cell's signaling cluster identity. However, this argument should be attenuated when taking Figure 2C into consideration. At least there is no simple linear relationship. Differences in the abundance of dividing cells become substantial already 10h post-stimulation. Here, cluster 3 exhibits the highest fraction of dividing cells and 5/6 the lowest with cluster 4 somewhere in between. Comparing this with the dynamics of the signaling clusters until 10h reveals something interesting. Cluster 3 has by far the highest amplitude and until 8h its median cytoplasmic/nuclear ratio does not drop below that of signaling cluster 4. This argues that the phenotypic progression is either related in a non-intuitive manner to translocation dynamics or that other, undescribed distinct cellular states (e.g. not characterized phenotypic features) determine the likelihood of cell division and influence the dynamics of SMAD2. Moreover, all signaling clusters exhibit a substantial fraction of dividing cells indicating that the link between signaling and the examined phenotypic response, while correlated, is weak at best.

The model of TGF- β signaling contains an abundance of kinetic rates and variables. However, for parameter estimation only very few measurements are used. Is the statistical power to identify parameters sufficient? The authors should elaborate on this shortly, and argue why this does not affect any of the conclusions made.

Populations stimulated with 5pM of TGF- β still contain a substantial fraction of non-responders. The model predicts an even higher fraction of non-responders for higher doses of TGF- β . Could the authors quickly elaborate on cellular characteristics which lead to this phenomenon?

Reviewer #2:

Strasen et al present an interesting work that investigates the sources of cellular response variability to TGF β as monitored by SMAD pathway. The authors created reporter cell line and used it to monitor SMAD2 translocation to the nucleus. They found that cellular response is variable and were able to cluster the cells into six classes. Computational modeling was used to recreate the observed variability in silico when using complex noise scheme. Finally, their model results pointed to a cluster-specific role of SMAD7 feedback that was verified experimentally.

Overall, I found the paper to be interesting and after the points below addressed, should be published in MSB. My specific comments are:

1. The cell line used is adequate and the controls (Figure 1+S1) indicate that the response is physiological. However, it is somewhat different from what was previously reported. It is not fully clear why that is.
2. The data analysis follows standard practices. However, there seem to be high levels of noise in the raw traces for SMAD2 and SMAD4 (Fig S1C-E + S2E-G). It is hard to know to what degree this matter (it might not) but given the nonlinear nature of the similarity measure used (cDTW) I just can't say. The authors need to do a better job addressing the question of experimental noise in the context of the cDTW measure.
3. The use of cDTW for similarity measure seems promising, however, the relaxation of time linearity make biological interpretation of similarity to be problematic. It is impossible for be to evaluate what a 0.25 malus means. For example, how different can two peaks be in terms of the residence time of a TF in the nucleus to still be mapped to each other? How different "feature based" metric, i.e. area under the curve, can be between two cells to still allow them cells to be perfectly aligned with each other?
4. Clustering was done based on a highly non-linear similarity metric (cDTW). So the fact that there is overall more similarity between clusters compared to similarity based on dose is pretty much uninformative. Of course, cells within a cluster are more similar to each other than by dose. That's how their algorithm assigned clusters, by that similarity! Therefore I find the comparison done in Figure 2E+F to be somewhat uninformative.

5. The basic model fitting (to overall population average, Fig 4) does not address all question associated with fitting uncertainty. They fit a complex model and try to estimate ~50 parameters from a single curve. It is likely that there are many sets of parameters that are all equally good fit to the data. The authors should do more to address questions related to fitting uncertainty.

6. The idea that a single fit is not a good representative of the population is interesting and timely and separation to prior classes is an interesting step. It is clear from Figure 6D that feedback protein production has some explanatory power with relation to classes. However, this seems to be very minimal (~0.2 bits). How much of this is only due to shifting in a single cluster? It would be much more useful to get an estimate of the overlap in parameters in units that are easier to interpret like percentage of cells that show the difference or something like that.

7. While the authors emphasize the fact that ~30% inhibition in the model agrees with SMAD7^{-/-} at high concentrations, it is interesting that it does not agree at all for 1 and 2.5 pM TGFb. A lack of agreement between model and data could be very informative and should be addressed.

Minor comments:

1. Figure 2 and S4: distances should always be reported in micron units and not pixels.
2. Figure 6D, really unclear why authors are using entropy and not mutual information. They are exactly looking at separation of classes and in the text, they ask the reader to compare (i.e. subtract...) the entropy they show from
3. Missing citation to highly relevant recent work in MSB, Yao et al (pmid: 27979909). Single cell parameter fitting was done to show that in MCF10A cells, Ca²⁺ response is really a mixture of three main classes.

Response to Reviewer

Reviewer #1:

General remark:

... The approach used ... is nice but remains very hypothetical unless the identified parameters that are causative are experimentally proven to drive cell-to-cell variability. In this light, their SMAD7 knockout experiments are certainly laudable. Perhaps the very least the authors could do is speculate how in growing populations of genetically identical adherent cells that are exposed to identical conditions differences in SMAD7 expression levels could arise and that this is still an open question.

To address this general remark, we expanded our discussion of sources of protein level variation (pages 24/25) and specifically addressed modulators of SMAD7 expression that were discussed in the literature, including signalling pathways mediated by PKC and HGF as well as miRNAs.

Points of criticism:

*1. Figure S1G shows quite different dynamics and wider **distributions** for YFP-SMAD2 activity then for endogenous SMAD2. Is YFP-SMAD2 a bona fide reporter, or should the authors have chosen for fluorescent protein tagging of endogenous SMAD, for instance using CRISPR/Cas9? This is a crucial point to be able to derive meaningful conclusions from the analysis of cell-to-cell variability.*

In the current study, we focused on analysing the activity of the TGF β pathway. To this end, we chose a transgenic reporter based on exogenous expression of YFP-SMAD2 or YFP-SMAD4, as its level and localization is only dependent on its post-translational modification by TGF β RI (see Fig. 1G). Endogenous reporters, in contrast, may also be affected by other regulatory processes affecting SMAD2 or SMAD4, such as gene expression, splicing, mRNA stability or translation control. In further studies, it will be interesting to compare endogenous and transgenic reporters.

The comparison of YFP-SMAD2 and endogenous SMAD2 translocation (Fig. S1G) is limited by the corresponding measurement techniques: we compare a genetically encoded fluorophore with immunofluorescence based on primary and secondary antibody binding, which is restricted by the quality (affinity, specificity) of the antibodies (Waters, 2009). This leads to differences in dynamic range and background level; the necessary normalization procedures affect the resulting distributions as well. Moreover, measurements were performed in different cell lines (parental MCF10A vs. SMAD reporter lines). Considering these limitations, we believe that the data provided for SMAD2 and SMAD4 translocation (Fig. S1G and S2C) provide good evidence for YFP-SMAD2 being a faithful reporter of pathway activity despite small differences in the kinetics of initial activation and distribution width. To further validate the reporter, we again performed immunostaining of total SMAD2 and SMAD4, but this time only in the SMAD reporter cell line, and compared nuclear levels upon stimulation with corresponding YFP-SMAD signals in the same cell (new Fig. EV1G and Fig. S2D). Despite expected technical limitations that increase noise such as loss of YFP fluorescence due to fixation, both signals were highly correlated with Pearson's correlation coefficients of 0.8 to 0.9.

*2. The authors use a defined, non-adaptive region in the **cytoplasm** which is solely dependent on the nuclear segmentation to quantify the cytoplasmic signal of YFP-SMAD2 and use the intensity in this region as a proxy for the whole cytoplasmic intensity. This is commonly used for proteins which exert their function by nuclear translocation. However, in figure 1D it seems that the cytoplasmic distribution of SMAD2-YFP is far from homogenous (E.g. 100 and 120 min). Thus, differences in the cytoplasmic/nuclear ratio between individual cells could be an artifact of localization differences in the cytoplasm. More critically, this might change as a function of cell area and shape which would raise major issues with respect of arguing that the cellular phenotype is not causative for the distinct signaling clusters. Is there any possible way of excluding this explanation?*

Due to the lack of a consistent cytoplasmic marker, we needed to estimate the cytoplasmic signal using a defined region around nuclear segmentations. These regions were grown in parallel from nuclear segmentations avoiding overlap with neighboring cytoplasmic and nuclear regions and thus adaptive to a certain extent. However, we could not exclude that they extend into extracellular regions. To minimize the influence of incomplete cytoplasmic segmentation, we determined the nuclear to cytoplasmic SMAD2 ratio using the median fluorescence intensity of the estimated cytoplasmic region for each cell over time, which is more robust against fluctuations than the mean intensity. We compared this measure to corresponding ratios calculated using the first or third quartile (new Fig. S1A). While absolute values varied as expected, all measures were linearly correlated with correlation coefficients above 0.95, indicating that ratios are robust against variations in cytoplasmic fluorescence intensities. We further analyzed whether the size of the cytoplasmic region influenced nuc/cyt SMAD ratios by measuring how they changed with a decreased width of the annulus (new Fig. S1B). Deviations were minimal for untreated cells and increased only slightly at higher stimulus levels due to the larger dynamic range of measurements. We then globally analyzed the normalized deviation of nuc/cyt SMAD ratios for increased and decreased widths of the cytoplasmic annulus and observed changes of less than 1% (new Fig. S1C). Taken together, these analyses demonstrate that our image analysis strategy provides robust measures of the nuc/cyt SMAD ratio and therefore of pathway activity in single cells. We now included this detailed discussion in the Appendix.

3. In figure 1B two distinct bands are visible for YFP-SMAD2 and only one can be observed for endogenous SMAD2. Could the authors elaborate on why this is the case? It is of interest since if there are two distinct forms of YFP-SMAD2 present in the population, the observed single-cell signaling response clusters might represent distinct mixtures of these two YFP-SMAD2 isoforms in each cell.

We regularly observed secondary bands both for YFP-SMAD2 and endogenous SMAD2. The 4-12% gradient gels used in the initial submission do not resolve the second band for endogenous SMAD2 very well and often exaggerate the upper band of YFP-SMAD2. We now repeated the western blot several times using more conventional 10% SDS polyacrylamide gels with linear size resolution. These show one main band for YFP-SMAD2 and endogenous SMAD2 as well as a light secondary band for both (updated Fig. 1B and below, biological replicate). Note that the secondary band for SMAD2 is also observed in the parental cell line, arguing that it reflects an endogenous variant or modification. Taken together, we are confident that there is only one form of YFP-SMAD2 present in the population and that heterogeneous single cell responses cannot simply be explained by distinct mixtures of YFP-SMAD2 isoforms.

4. Figure 1C shows dynamics of phosphorylation of SMAD2 and YFP-SMAD2 and the authors argue that the dynamics are comparable. Judging from the blots it appears that pYFP-SMAD2 decreases substantially faster after stimulation than its endogenous counterpart. A quantification would be

helpful. Moreover, one sees a double band also for pYFP-SMAD2 but the upper band seems to be less phosphorylated than the lower band. What does that mean?

We repeated western blot analysis of SMAD2 activation using 10% SDS polyacrylamide gels (see above) and quantified phospho-SMAD2 and phospho-YFP-SMAD2 levels in parental and reporter cell lines (updated Fig. 1C and new Fig. EV1A, 4 biological replications). We observed that phosphorylation dynamics of SMAD2 and YFP-SMAD2 were comparable within the error range. Due to its lower expression levels, YFP-SMAD2 phosphorylation could not be detected at early time points (15 min post stimulus), which explain the apparent lag phase. All phosphorylated SMAD2 forms decayed with similar kinetics. We did not observe phosphorylation of secondary SMAD2 bands (see above) probably due to their low levels. Taken together, we believe that quantification of several western blots confirms that YFP-SMAD2 and endogenous SMAD2 are phosphorylated with similar kinetics.

*5. The authors extensively describe how individual cells are tracked, segmented and signal is quantified. However, there is no description of how **mitotic cells** are treated with respect to assigning them to the signaling classes. This is especially relevant in figure 1F. E.g. when a cell divides at five hours after stimulation how can a distinction between classes two and four be made which look remarkably similar until that point. Are the offspring's dynamics included and merged with its mother's dynamics?*

For most analyses, we track cells in forward direction from the first to the last time point. When a cell divides, only the daughter cell closest to the last position of the mother is followed, whereas the other daughter is excluded from further analyses. Data from mothers and their closest offspring are merged and only trajectories of the same length covering the full observation period, which corresponds to 24h / 289 time points in most cases, are considered. Given that signalling classes were derived from trajectories spanning the full time course, we were able to assign trajectories to defined signalling classes based on the offspring, even if mothers divided before classes can be distinguished.

For sister cell analyses (Fig. 3E), cells are tracked backwards from the last to the first time point. Again, data from offspring and mothers are merged and we only consider full-length trajectories. As a consequence, trajectories of sister cells are identical before cell division. During mitosis itself, nuclear envelope breakdown prevents meaningful measurements of SMAD translocation, thus we interpolate corresponding values. To clarify how we handle cell division during tracking and data analysis, we expanded the corresponding paragraph in the methods sections and the appendix.

*6. Figure 2F tries to exemplify that the fraction of **dividing cells**, i.e. in their argument the phenotypic response of individual cells, is determined by a **cell's signaling cluster** identity. However, this argument should be attenuated when taking Figure 2C into consideration. At least there is no simple linear relationship. Differences in the abundance of dividing cells become substantial already 10h post-stimulation. Here, cluster 3 exhibits the highest fraction of dividing cells and 5/6 the lowest with cluster 4 somewhere in between. Comparing this with the dynamics of the signaling clusters until 10h reveals something interesting. Cluster 3 has by far the highest amplitude and until 8h its median cytoplasmic/nuclear ratio does not drop below that of signaling cluster 4. This argues that the phenotypic progression is either related in a non-intuitive manner to translocation dynamics or that other, undescribed distinct cellular states (e.g. not characterized phenotypic features) determine the likelihood of cell division and influence the dynamics of SMAD2. Moreover, all signaling clusters exhibit a substantial fraction of dividing cells indicating that the link between signaling and the examined phenotypic response, while correlated, is weak at best.*

TGF β signaling interferes with cell cycle progression through various molecular mechanisms such as expression of the CDK inhibitor p21 or downregulation of the phosphatase CDC25 (Derynck *et al*, 2001). However, there are also mechanisms like Myc expression that counteract TGF β mediated growth inhibition. As a consequence, TGF β signaling does not lead to a complete cell cycle arrest in many epithelial cells, but rather induces delayed S-phase entry and longer cell cycle duration. We see similar effects in MCF10A cells under our culturing conditions, which include significant amounts of EGF and Insulin as mitogenic stimuli: cells still divide after stimulation with TGF β , but cell cycle length is significantly increased. Nonetheless, we agree with the reviewer that the observed effect is not very strong and toned down our discussion accordingly.

As we measured cell division from time-resolved single-cell measurements, we showed the corresponding data as cumulative sums over time. This gave the impression that we intended to directly compare signaling dynamics with cell cycle progression at all time points. However, at early time points, only few cells divided, limiting the power to distinguish between groups. We therefore believe that it is difficult to draw conclusions about the relationship of amplitudes at intermediate time periods (8-10h) and cell cycle progression. It is more robust to compare the (integrated) effect on cell cycle progression at the end of the observation period (24h). To avoid misconceptions, we altered our presentation of cell division and now show fraction of cells dividing once or twice or not at all during the 24h period post stimulation in relation to signaling classes or stimulus levels (updated Fig. 2F).

7. The model of TGF- β signaling contains an abundance of kinetic rates and variables. However, for parameter estimation only very few measurements are used. Is the statistical power to identify parameters sufficient? The authors should elaborate on this shortly, and argue why this does not affect any of the conclusions made.

To verify parameter estimation, we analysed whether their values are identifiable based on the data used for fitting. We did this by comparing parameter values of 30 independent model fits obtained from local multi-start optimization, which explained the data similarly well (new Fig. S4). We concluded that only few parameters are confined to narrow ranges when fitting our model to the population-average dataset, whereas the remainder can vary substantially despite a similar goodness-of-fit.

Despite non-identifiability of many parameter values, the population-average models nevertheless generated robust predictions for previously untested experimental conditions, which in our opinion is more important than a precise identification of each individual parameter value. To emphasize this, we now show model predictions of the top 30 fits obtained from local multi-start optimization for ligand degradation (updated Fig. 4D), restimulation experiments (updated Fig. 4E-G) and DRB treatment (updated Fig. 4H). In each case, we observed that predictions of all measurement-compliant parameter sets were confined to a narrow range.

We similarly addressed the robustness of our model predictions concerning cell population heterogeneity. To this end, we independently fitted the six signalling classes from 30 independent population-average fits allowing only protein concentrations to change. Based on these 30 fits, we generated artificial cell populations by adding the same protein concentration noise as in the best-fit model and assessed the robustness of our predictions for SMAD7 knockout. In all fits, we observed similar transitions from transient to sustained signalling classes upon loss of SMAD7 at high TGF β concentrations (new Fig. EV5D, upper panel). At low TGF β concentrations, we mainly observed a decrease in non-responding cells upon SMAD7 knockout, whereas the transient signalling classes persisted (new Fig. EV5D, lower panel). Hence, each of these models predicted that feedback is a major signal adaptation mechanism at high doses, whereas ligand degradation dominates signal termination upon weak stimulation.

Taken together, we think that the model is capable of robustly generating experimentally testable predictions, even though individual parameter values themselves are not completely identifiable. We believe that this is mainly due to the fact that the ratio of certain parameters rather than their absolute values determines SMAD translocation dynamics as illustrated in our response to question 8. We now discuss the analysis of identifiability in the appendix and refer to it in the revised results section. We further highlight that we used not only the nuclear to cytoplasmic SMAD2 ratio for calibrating the population-average model, but considered a substantial amount of experimental data including time-resolved measurements of SMAD4 and receptor levels, perturbation experiments using small-molecule inhibitors as well as restimulation experiments (Figure EV4 and Table S4).

8. Populations stimulated with 5pM of TGFβ still contain a substantial fraction of non-responders. *The model predicts an even higher fraction of non-responders for higher doses of TGFβ. Could the authors quickly elaborate on cellular characteristics which lead to this phenomenon?*

To characterize non-responding cells, we took advantage of the top 30 fits to the signalling classes (see response to point 7 of this reviewer) and asked whether the fits to class 1 (non-responders) were characterized by a different set of protein concentrations as opposed to the other classes (responders). We found that class 1 fits tend to harbour less TGFβ receptors and more feedback regulators than the remaining signalling classes (new Fig. S5A). Accordingly, the most predictive feature distinguishing responders from non-responders was the ratio of feedback regulator expression to the sum of TGFβRI and TGFβRII expression levels, further underscoring the importance of parameter ratios for determining SMAD dynamics (new Fig. S5B).

The analysis above used fits to median dynamics of signalling classes at 100 pM TGFβ, but did not contain direct information about the behaviour at 5 pM TGFβ, where non-responders are much more prevalent. To analyse this regime, we used the corresponding simulated heterogeneous cell population mapped onto the signalling class dynamics (Fig. 6A). When plotting signalling protein concentration distributions over all cells in class 1-6, we again observed that the feedback-to-receptor ratio is a good predictor that discriminates responders (class 2-5) vs. non-responders (class 1) (new Fig. S6).

While the above results merely show that the feedback-to-receptor ratio correlates with the emergence of non-responders, we also provide direct proof for a causal relationship in our manuscript, as SMAD7 knockout decreases the fraction of non-responders over multiple TGFβ doses, in both model and data. However, as pointed out by the reviewer the absolute number of non-responders is different between model and data, and we mention this discrepancy in the revised manuscript. We now discuss characteristics of non-responding cells in the context of deviating model predictions to SMAD7 knock-out (see response to point 7 of reviewer 2). We furthermore added the above-mentioned results concerning the feedback-to-receptor ratio to the main text, and added corresponding new supplementary figures (new Fig. S5 and S6).

Reviewer #2:

1. *The cell line used is adequate and the controls (Figure 1+S1) indicate that the response is physiological. However, it is somewhat **different from what was previously reported**. It is not fully clear why that is.*

The average response of our single cell measurements corresponds well with biochemical measurements in previous studies (Vizan *et al*, 2013; Zi *et al*, 2011; Inman *et al*, 2002; Clarke *et al*, 2009). However, our results conflict with a previous study that used time-lapse microscopy of mouse myoblast C2C12 cells (Warmflash *et al*, 2012) and reported sustained SMAD2 and transient SMAD4

nuclear accumulation upon TGF β stimuli. Potential explanations for the observed discrepancies are cell-type and species-specific effects as well as different experimental setups, as the reporter is expressed to about half the level of endogenous SMAD2 in our study in contrast to >2-fold overexpression in C2C12 cells. We address these issues in the discussion section on page 21 and adjusted the corresponding paragraph in the results section as well.

2. *The data analysis follows standard practices. However, there seem to be high levels of noise in the raw traces for SMAD2 and SMAD4 (Fig S1C-E + S2E-G). It is hard to know to what degree this matter (it might not) but given the nonlinear nature of the similarity measure used (cDTW) I just can't say. The authors need to do a better job addressing the question of **experimental noise in the context of the cDTW measure.***

To analyze the robustness of the cDTW measure, we added white noise to measured single cell trajectories and calculated cDTW distances to the unaltered time series. We then normalized the resulting cDTW distances between noisy and unaltered trajectories to the average distances between the experimentally observed signalling classes (new Fig. S3D). From this analysis, we conclude that experimental noise will have only a minor effect on cDTW-based clustering, as even high signal-to-noise ratios of up to 25% led to cDTW distances between noisy and unaltered trajectories that correspond to only around 10% of the mean distances between the observed signaling classes. We included this discussion on experimental noise in a revised analysis of cDTW performance in the appendix and refer to it in the main text.

3. *The use of cDTW for similarity measure seems promising, however, the relaxation of time linearity makes biological interpretation of similarity to be problematic. It is impossible for me to evaluate what a **0.25 malus** means. For example, how different can two peaks be in terms of the residence time of a TF in the nucleus to still be mapped to each other? How different "feature based" metric, i.e. area under the curve, can be between two cells to still allow them cells to be perfectly aligned with each other?*

To provide a better understanding of the advantages and limitation of cDTW, we generated an artificial noisy time course with a single peak, systematically altered it by shifting or stretching and compared resulting cDTW scores with varying strength of the parameter *malus* to the Euclidian distance (new Fig. S3A-B). This analysis indicated that without constrain (*malus* = 0), even temporally distant or highly stretched peaks were aligned, leading to low cDTW scores. Increasing *malus* restricted the ability of the algorithm to align peaks. When using a *malus* of 0.25, peaks shifted by about 300min were no longer recognized as similar and cDTW scores approached Euclidian distance. Similar effects were observed for stretched peaks, although on slightly longer time scales. However, even though distant peaks could be aligned in the artificial data, the cDTW scores always increased with increasing shift or stretch. In addition to constraining DTW by the parameter *malus*, we also employed a Sakoe-Chiba band during calculation (see Appendix), which restricts alignments to time points that are no more than 125 minutes apart. As a result, we observed a "time warp" of less than 75min for most data points (mean: 46 minutes) in the real data set used to identify signaling classes (new Fig. S3C), suggesting stretching and shifting is actually limited in our signaling dynamics classification. We included this discussion in a revised analysis of cDTW performance in the appendix and refer to it in the main text.

4. *Clustering was done based on a highly non-linear similarity metric (cDTW). So the fact that there is overall more similarity between clusters compared to similarity based on dose is pretty much **uninformative.** Of course, cells within a cluster are more similar to each other than by dose. That's how their algorithm assigned clusters, by that similarity! Therefore, I find the comparison done in Figure 2E+F to be somewhat uninformative.*

We used the agglomerative Ward's method for clustering, which minimizes the within-cluster variance based on the sum of squares between two clusters, and refined it with a centroid model based algorithm utilizing the Hausdorff distance. We wanted to visualize the quality of the clustering and therefore show results from applying the silhouette method on the grouped data. The silhouette method measures the intracluster divergence (tightness) compared to intercluster divergence (separation) and can be applied to clusters that were generated by many different algorithms. It provides an intuitive visual confirmation that clusters based on SMAD dynamics provide a different separation of single cell responses than grouping cells according to ligand dose. However, we do not want to overstate the significance of this analysis and adjusted the corresponding section in the main text.

*5. The basic model fitting (to overall population average, Fig 4) does not address all question associated with fitting uncertainty. They fit a complex model and try to estimate ~50 parameters from a single curve. It is likely that there are many sets of parameters that are all equally good fit to the data. The authors should do more to address **questions related to fitting uncertainty**.*

see Reviewer #1, point 5

*6. The idea that a single fit is not a good representative of the population is interesting and timely and separation to prior classes is an interesting step. It is clear from Figure 6D that **feedback protein** production has some explanatory power with relation to classes. However, this seems to be very minimal (~0.2 bits). How much of this is only due to **shifting in a single cluster**? It would be much more useful to get an estimate of the overlap in parameters in units that are easier to interpret like percentage of cells that show the difference or something like that.*

To address this comment, we now show the signalling protein distributions in the top 30 fits for each of the six signalling clusters (see Fig S5A). This is the primary data underlying the entropy calculation, and thus a more easily interpretable quantity, as requested by the reviewer. However, it turns out that these protein distributions are very complex unless parameter combinations like the feedback-to-receptor ratio are considered (see response to comment 8 of reviewer 1). Specifically, no particular protein or cluster clearly stands out, suggesting that the explanatory power of individual proteins is limited as commented by the reviewer. In fact, in the initial manuscript we first analysed the effect of feedback depletion in the model, and only then additionally stated that the level of the feedback regulator also distinguishes signalling classes according to our entropy calculation as a second line of evidence. We decided to keep the entropy figure in the revised manuscript and formulated our conclusions more carefully.

7. While the authors emphasize the fact that ~30% inhibition in the model agrees with SMAD7^{-/-} at high concentrations, it is interesting that it does not agree at all for 1 and 2.5 pM TGF β . A lack of agreement between model and data could be very informative and should be addressed.

Model predictions for SMAD7 knock-out (Fig. 6G) and experimental measurements (Fig. 6E, right) deviate at 1 and 2.5pM TGF β mainly due to the model underestimating the fraction of non-responding cells. As described in our response to point 8 of reviewer 1, we further analysed which parameters characterize the class of non-responding cells in our independent model fits and observed that non-responding cells are best characterized by the ratio of feedback to receptor levels (new Fig. S4B and Fig S5). This suggests that cells could compensate for the loss of SMAD7 by downregulating receptor levels. To support this hypothesis, we measured receptor levels in knock-out cells by western blot and observed a substantial decrease of TGF β RII (new Fig. S7).

It is also worth mentioning that the model was successfully calibrated using an initial dataset (Fig. 2D) that has a lower number of non-responders compared to the parental control of the SMAD7

knockout (Fig. 6E, right). This explains why the unperturbed model (Fig. 6A) already has less non-responders than the parental control data shown in Fig. 6E. Hence, there is a systematic batch effect between initial data and model vs. subsequent SMAD7 knockout data reflecting for example differences in cell lines, cell density or effective TGF β concentration. We therefore think that the deviation between model and new data in absolute terms is tolerable and it is more important that the model robustly reflects shifts in the signalling classes upon SMAD7 knockout such as the loss of classes 1-3 at high doses or the loss of non-responders at low doses (new Fig. EV5D). Finally, our presentation of the results was previously misleading, as the model prediction (Fig. 6G; 1-100 pM) had less columns than the experimental measurements (Fig. 6E; 0-100 pM), which made the visual comparison of the two difficult. We eliminated this confusing factor by removing the column representing unstimulated cells from the experimental data, but note that a deviation between model and data persists.

In addition to addressing the deviation in non-responding cells in the main text, we expanded our discussion why SMAD7 knockout has differential impact on the signaling classes depending on the applied TGF β dose (page 25).

Minor comments:

1. *Figure 2 and S4: distances should always be reported in micron units and not pixels.*

We converted units from pixel to micron in Fig. 2G and Fig. EV2F-I.

2. *Figure 6D, really unclear why authors are using entropy and not mutual information. They are exactly looking at separation of classes and in the text, they ask the reader to compare (i.e. subtract...) the entropy they show from*

We chose entropy for Fig. 6D to emphasize the heterogeneity of each parameter among the signaling classes in a given model fit. It is a simpler measure compared to mutual information, which would require further normalization and discretization to calculate conditional entropies for parameter changes and signaling classes. Our rationale is that the parameter with the highest information content (lowest entropy) might be a good candidate for further studies. While we agree with the reviewer that using mutual information would be as well a suitable way to analyze which parameters contribute most to the separation of signaling classes, we would prefer to present entropy in the context of this manuscript.

3. *Missing citation to highly relevant recent work in MSB, Yao et al (pmid: 27979909). Single cell parameter fitting was done to show that in MCF10A cells, Ca²⁺ response is really a mixture of three main classes.*

We apologize for missing to cite this study, which is indeed relevant to our work. We now mention it in the discussion section in the context of the decomposition of the TGF β response into distinct signalling classes (page 21/22) as well as in the context of model fitting to single cell data (page 22).

References:

Clarke DC, Brown ML, Erickson RA, Shi Y & Liu X (2009) Transforming growth factor beta depletion is the primary determinant of Smad signaling kinetics. *Mol Cell Biol* **29**: 2443–2455

Derynck R, Akhurst RJ & Balmain A (2001) TGF- β signaling in tumor suppression and cancer progression. *Nat Genet* **29**: 117–129

Inman GJ, Nicolás FJ & Hill CS (2002) Nucleocytoplasmic shuttling of Smads 2, 3, and 4 permits sensing of TGF-beta receptor activity. *Mol Cell* **10**: 283–294

Vizan P, Miller DSJ, Gori I, Das D, Schmierer B & Hill CS (2013) Controlling Long-Term Signaling: Receptor Dynamics Determine Attenuation and Refractory Behavior of the TGF- Pathway. *Science Signaling* **6**: ra106–ra106

Warmflash A, Zhang Q, Sorre B, Vonica A, Siggia ED & Brivanlou AH (2012) Dynamics of TGF-signaling reveal adaptive and pulsatile behaviors reflected in the nuclear localization of transcription factor Smad4. *Proc Natl Acad Sci USA* **109**: E1947–E1956

Waters JC (2009) Accuracy and precision in quantitative fluorescence microscopy. *J Cell Biol* **185**: 1135–1148

Zi Z, Feng Z, Chapnick DA, Dahl M, Deng D, Klipp E, Moustakas A & Liu X (2011) Quantitative analysis of transient and sustained transforming growth factor- β signaling dynamics. *Mol Syst Biol* **7**: 492

Thank you again for submitting your work to Molecular Systems Biology. We have now heard back from the referee who agreed to evaluate your study. As you will see below, the reviewer is satisfied with the modifications made and thinks that the study is now suitable for publication.

Before we formally accept your manuscript for publication, we would ask you to address the following editorial issues:

- We have implemented a "model curation service" for papers that contain mathematical models. This is done together with Prof. Jacky Snoep and the FAIRDOM team and it is still in a pilot phase (therefore has not yet been announced officially). In brief, the aim is to enhance reproducibility and add value to papers containing mathematical models. Jacky Snoep's summary on the model (*Model Curation Report*) is pasted below. As you are already aware from your email exchange with him, and will also see in the report below, there are some minor issues, which we would ask you to fix when you submit your revision.

REVIEWER REPORTS

Reviewer #2:

The authors have nicely addressed all the comments I had. I recommend that the paper will be published in MSB in its current form.

Model Curation Report

The mathematical models described in the manuscript were uploaded to Biomodels, but were not publicly available, so we contacted the authors who sent us the model files in SBML format. The 12 models were used for the simulations shown in Figures 4 and 5 in the manuscript. The authors gave detailed instructions on how to reproduce the model simulations shown in the figures. Since the SBML files were constructed in Copasi we used the same simulator to test the reproducibility of the simulations. Although we could approach the results, our simulations were not identical to the results shown in the manuscript. In several iterations, we could sort out the problems that caused the differences between the simulations obtained with the submitted model files and the simulations shown in the manuscript. These problems were related to an incorrect translation of the model files from Matlab into SBML format, mostly with respect to formulation of the initial conditions. The authors corrected the initial conditions and wrong assignments of parameter values, and submitted corrected SBML models to us. With these models we could reproduce most of the simulation figures in the manuscript accurately. The authors noted that one of the figures in the manuscript is incorrect and will submit a corrected file for a next version of the manuscript.

The only small issue that could not be completely resolved was the simulation for a 1pM concentration in Fig. 4C. The small difference is due to the specific numerical integration routine and injection function used (as explained in the supplementary material). I am satisfied with the explanation given by the authors for this small difference (in second decimal) in the simulation results.

The updated SBML files and SED-ML scripts for the model simulations will be made available if the manuscript is accepted for publication.

Below are two actions listed that should be taken by the authors if a new version is uploaded. The authors are aware of this, and the actions are listed here solely as a reminder.

- 1) the authors should upload a new set of corrected SBML models
- 2) the authors should correct Figure 4D in the manuscript

We have addressed the remaining editorial issues and the comments from Model Curation Report as outlined below:

1. We have uploaded a new set of corrected SBML models to the BioModels database and adjusted the corresponding identifiers in the Data Availability section (MODEL1712050001 -MODEL17120500012).
2. We corrected Figure 4D.

Corresponding Author Name: Alexander Loewer

Manuscript Number: MSB-17-7733